# T-cell protrusions enable fast, localised initiation of chimeric antigen receptor signalling

Carmen Rodilla-Ramirez [1✉], Giorgia Carai[1], Eleanor Fox [1,3], Amin Zehtabian[1], Helen Adam[1], Katja Dallio[1], Pia Lazki-Hagenbach[1], Helge Ewers [1], Xiaolei Su [2] & Francesca Bottanelli [1✉]

## Abstract

**Actin-rich protrusions densely cover the surface of T cells and are well characterised for their role in migration. Recent studies have uncovered their contribution to antigen surveillance and immune signalling. To further explore how protrusions initiate signalling pathways mediating T-cell activation, we performed live-cell imaging of endogenously tagged proteins in HER2-specific chimeric antigen receptor (CAR) T cells targeting HER2+ breast-cancer cells. Quantitative STED microscopy allowed us to monitor protein rearrangement and to correlate it with membrane topology over time. Before activation, key signalling proteins (including Lck, CD45, LAT, and the CAR) were not enriched in protrusions. Upon contact with target cells, rapid protein reorganisation occurred preferentially within protrusions, initiating signalling. HER2-CAR clustering, accompanied by ZAP-70 and LAT recruitment, was enhanced in protrusions. While Lck distribution remained unchanged, exclusion of the phosphatase CD45 was enhanced at protrusion-cell contacts, independently of the CAR signalling domain. Overall, signalling machinery rearranged faster and more effectively at protrusive contacts than at main plasma membrane regions. Together, our data re-frame protrusions as sites of enhanced receptor activation by exclusion and clustering dynamics rather than by pre-enrichment of the signalling machinery.**

Subject Categories Cell Adhesion, Polarity & Cytoskeleton; Immunology; Signal Transduction

## Introduction

T lymphocytes play a central role in the adaptive immune response, using their T-cell receptor (TCR) to recognise foreign antigen peptides presented by major histocompatibility complexes (MHCs) on the surface of antigen-presenting cells (APCs). Upon sustained TCR interaction with peptide MHC (pMHC), the Src family kinase Lck phosphorylates immune receptor tyrosine-based activation motifs (ITAMs) on the cytoplasmic domains of the TCR, creating docking sites for the kinase ZAP-70 (Iwashima et al, 1994). Once at the TCR, ZAP-70 is activated by Lck and subsequently phosphorylates substrates such as the linker for activation of T cells (LAT) (Zhang et al, 1998). LAT then scaffolds a signalling hub for downstream signal propagation (Balagopalan et al, 2013; Balagopalan et al, 2015; Su et al, 2016). Researchers have exploited the power of T cells to target tumours that evade the immune system by arming T cells with Chimeric Antigen Receptors (CARs). CARs are synthetic receptors that typically combine antibody-derived ligand-binding motifs with intracellular domains of the TCR and coreceptors (Xiong et al, 2024). CAR signalling harnesses proximal effector molecules similar to those of the TCR, including Src-family kinases, ZAP-70 or LAT, but differs in magnitude and kinetics (Davenport et al, 2018; Gudipati et al, 2020; Karlsson et al, 2015; Salter et al, 2021).

Beyond the molecular components of T-cell signalling, T-cell function is also influenced by its complex membrane topography. The T-cell surface is covered by a heterogenous set of actin-rich protrusions (Alexander et al, 1976; Kenney et al, 1986; Majstoravich et al, 2004) that have traditionally been studied in the context of T-cell migration (Carman et al, 2007; Caswell and Zech, 2018; Dupré et al, 2015; Song et al, 2014). Studies over the past decade have highlighted a crucial role of actin-rich protrusions in immune signalling (Aramesh et al, 2021; Beppler et al, 2023; Cai et al, 2022; Cai et al, 2017; Jenkins et al, 2023; Kim et al, 2018; Orbach and Su, 2020; Sage et al, 2012; Tamzalit et al, 2019). Actin-rich protrusions, commonly referred to as T-cell microvilli, have been shown to rapidly scan the APC surfaces and stabilise upon TCR-pMHC binding (Cai et al, 2017) or hyper-stabilise upon CAR-ligand interaction (Beppler et al, 2023). Importantly, actin-rich protrusions are thought to be the preferred energetic structure for penetrating the glycocalyx of both T cell and target cell (Göhring et al, 2022; Pettmann et al, 2018), which can physically hinder intercellular receptor-ligand interactions (Ardman et al, 1992; Bell et al, 1984). Protrusions have been shown to "punch" into the target cell and bring T cell and target membranes closer to allow surface receptor interactions (Sage et al, 2012; Sanderson and Glauert, 1979). Disruption of the T-cell actin cytoskeleton delays this close contact formation (Jenkins et al, 2023), suggesting the need for actin-driven forces for penetration of the glycocalyx. An abundant

[1]Institute of Chemistry and Biochemistry, Freie Universität Berlin, Berlin, Germany. [2]Department of Cell Biology, Yale School of Medicine, New Haven, CT, USA. [3]Present address: Cambridge Institute for Medical Research, University of Cambridge, Cambridge, UK. ✉E-mail: carmenair97@zedat.fu-berlin.de; bottanelli@zedat.fu-berlin.de

component of the T-cell glycocalyx is the large phosphatase CD45. Size-dependent exclusion of CD45 from TCR-pMHC or CAR-ligand interaction sites, shown at different timepoints of T-cell activation (Chang et al, 2016; Razvag et al, 2018; Varma et al, 2006; Xiao et al, 2022), has been proposed to be the driving element for TCR and CAR triggering, according to the kinetic segregation model (Davis and van der Merwe, 2006).

A quantitative dynamic framework describing macromolecular rearrangements in relation to membrane topography on the surface of T cells, from the early contact to immunological synapse formation and in an unperturbed living cell, is currently lacking. The localisation of signalling proteins has conventionally been imaged by overexpressing proteins of interest fused to a fluorescent tag (Gudipati et al, 2020; Razvag et al, 2018) or via immunostaining of fixed cells (Beppler et al, 2023; Dong et al, 2020; Ghosh et al, 2020; Jung et al, 2016; Jung et al, 2021). While fixation has been instrumental for high-resolution imaging, it may alter membrane structures (Ichikawa et al, 2022; Schnell et al, 2012; Wong-Dilworth et al, 2023), underscoring the importance of complementary live-cell imaging techniques. Single molecule localisation microscopy (SMLM) has been broadly employed to assess the nanoscale localisation and clustering behaviour of signalling molecules on the plasma membrane of T cells, offering powerful insights (Lillemeier et al, 2010; Sherman et al, 2011; Williamson et al, 2011), though careful controls are needed to account for potential clustering artefacts (Rossboth et al, 2018).

The localisation of signalling molecules in T-cell protrusions prior to activation has been an active area of investigation, due to the possibility that T cells may enrich signalling molecules at their tips to facilitate initial antigen sensing. While some studies report enrichment of TCR and pre-exclusion of CD45 in protrusions in resting T cells (Ghosh et al, 2020; Jung et al, 2016; Jung et al, 2021), others present contrasting findings (Cai et al, 2022), reflecting the complexity and dynamic nature of these structures. Furthermore, most studies on actin-rich protrusions have relied on imaging T-cell interactions with coated glass surfaces (Chang et al, 2016; Razvag et al, 2018) or lipid bilayers (Beppler et al, 2023; Cai et al, 2022; Cai et al, 2017; Jenkins et al, 2023), where membrane topology in relationship to domains enrichment is hard to infer due to the lack of axial resolution in most conventional and super-resolution imaging methods. While this system has greatly advanced our understanding of T-cell signalling, it highlights the need for approaches that capture the complex topography of the T-cell plasma membrane during more physiological cell–cell interactions.

Here, we employed CRISPR-Cas9 to endogenously tag key signalling proteins such as Lck, ZAP-70, LAT or CD45, and performed quantitative live-cell confocal and super-resolution [stimulated emission depletion (STED)] microscopy to develop a quantitative model of their dynamics and nanoscale redistribution during Human Epidermal Receptor 2 (HER2)-CAR-mediated activation at cell–cell contacts. CAR systems not only provide a powerful model to study early cell–cell interactions and the role of membrane topology and protrusions in signalling, but also offer insights that could guide the improved engineering of CAR T cells for more effective cancer targeting (Xiong et al, 2024). Our findings indicate that the HER2-CAR itself, LAT, Lck and CD45 do not localise preferentially to actin-rich protrusions prior to contact with a target cell. Within the first 2 min of target cell contact—preferentially initiated through actin-rich protrusions—CAR and LAT clustering, as well as ZAP-70 recruitment, are enhanced within these structures compared to other membrane regions. This is likely due to more efficient exclusion of CD45 from protrusions, as the kinase Lck shows no significant redistribution at the macroscale.

# Results

## Endogenously tagged signalling proteins are not enriched in actin-rich protrusions prior to T-cell activation

To understand the contribution of protrusions to early signalling events in CAR-mediated T-cell activation, we aimed to gain a better understanding of how key signalling proteins rearrange within T-cell protrusions. As a first step, we analysed their nanoscale localisation under resting conditions. Previous studies have reported contradictory results on the localisation of signalling proteins in protrusive structures prior to activation in fixed T cells (Cai et al, 2022; Ghosh et al, 2020; Jung et al, 2021). Those reporting enrichment of TCR, Lck and LAT in protrusions suggested it may favour activation, as protein confinement in microvilli may amplify early TCR signalling. We thus wanted to dissect the distribution of LAT, Lck and CD45 in actin-rich protrusions in an unperturbed living T cell using confocal and super-resolution STED microscopy, a purely optical imaging technique.

To avoid artefacts generated by protein overexpression, we knocked in a Halo tag at the endogenous loci of ZAP-70, Lck, LAT or CD45 in Jurkat T cells (Fig. 1A). The successful tagging of the proteins was verified via western blot (Fig. EV1), and the ability of the gene-edited knock-in (KI) cells to normally activate was assessed by measuring their IL-2 secretion (Fig. EV2A). In order to confirm the functionality of the tagged proteins, we confirmed that phosphorylation of LAT in the ZAP-70[EN]-Halo cell line (Fig. EV2B,C) and phosphorylation of CD3ζ in the Lck[EN]-Halo and the CD45[EN]-Halo cell lines (Fig. EV2D,E) were not affected. Additionally, the overall increase in phosphotyrosine levels upon activation in the LAT[EN]-Halo cell line did not differ from that observed in Jurkat WT cells (Fig. EV2F–G). Our results are consistent with previous reports showing that C-terminal tagging of Lck (Ehrlich et al, 2002), ZAP-70 (Sloan-Lancaster et al, 1998) and LAT (Saez et al, 2025) did not alter protein functionality. While ZAP-70 displayed cytoplasmic localisation prior to activation, live-cell STED imaging of LAT, Lck and CD45 highlighted a strong plasma membrane localisation of all proteins that marked both the main body membrane and protrusions, without displaying any striking enrichment in protrusions (Fig. 1A). Live-cell STED further highlighted local clustering of Halo-tagged Lck, LAT and CD45 at the plasma membrane, suggesting a certain degree of macromolecular assembly under resting conditions, consistent with previous reports (Lillemeier et al, 2010; Rossy et al, 2013; Sherman et al, 2011).

To confirm actin enrichment and to later analyse the enrichment of signalling proteins in T-cell protrusions, we developed an unbiased protrusion segmentation pipeline (Appendix Fig. 1A). This approach allowed us to quantify protein *enrichment* by normalising the average intensity within protrusions to the average

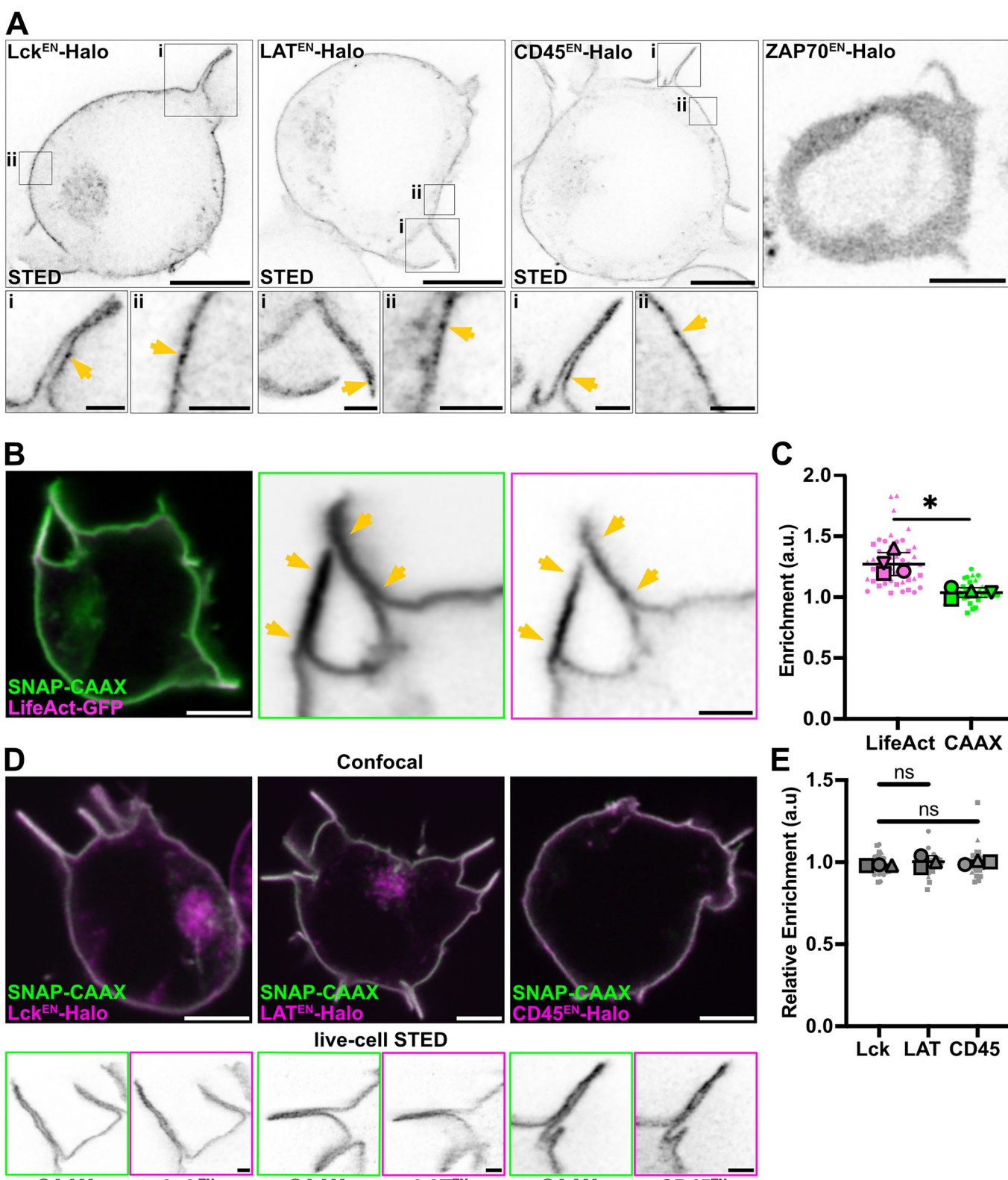

intensity across the full plasma membrane (Appendix Fig. 1B). To account for possible sampling artefacts, such as increased signal in protrusions due to a higher membrane-to-pixel ratio, we compared the distribution of each protein with a SNAP-tagged CAAX motif (Moores et al, 1991), which serves as an evenly distributed plasma membrane marker. We calculated each protein's *relative enrichment* by normalising its protrusion *enrichment* to that of the membrane marker in the same cell (Appendix Fig. 1B). A *relative enrichment* greater than 1 indicates enhanced localisation within protrusions, less than 1 indicates exclusion, and a value of 1 reflects

◄  **Figure 1.   Live-cell STED imaging of endogenously tagged proteins revealed that signalling proteins are not enriched in actin-rich protrusions prior to T-cell activation.**

(**A**) Live-cell confocal and STED imaging of Jurkat T cells expressing endogenously Halo-tagged ZAP-70, Lck, LAT and CD45 labelled with CA-JFX$_{650}$. Crops show protrusions (i) and main body membranes (ii). Arrows highlight the heterogeneous distribution of the proteins in the membrane. (**B**) Live-cell imaging of Jurkat T cells expressing LifeAct-GFP and SNAP-CAAX (labelled with BG-JFX$_{650}$). Arrows highlight protrusions enriched in F-actin. (**C**) *Enrichment* of LifeAct-GFP or SNAP-CAAX in protrusions. In total, 45 cells from four independent experiments were analysed. The graph shows mean values, standard deviation (s.d.) error bars. *P* value of paired *t* test is 0.0142. (**D**) Live-cell confocal (magenta and green) and STED images (inverted greyscale) of Jurkat T cells expressing Lck$^{EN}$-Halo, LAT$^{EN}$-Halo and CD45$^{EN}$-LAP-Halo labelled with CA-JFX$_{650}$ and SNAP-CAAX labelled with BG-JF$_{571}$. (**E**) *Relative enrichment* of LAT, Lck and CD45 in protrusions. In total, 32 cells (for Lck), 34 cells (for LAT) and 34 cells (for CD45) from three independent experiments were analysed. Replicates are shown as different shapes, and each small dot represents a single cell. The graph shows mean values and s.d. error bars. *P* values of unpaired *t* tests are 0.1411 (Lck/CD45) and 0.7628 (LAT/CD45). CA chloroalkane (HaloTag substrate), BG benzylguanine (SNAP-tag substrate). Scale bars, 5 μm (confocal overviews), 1 μm (crops and STED images).

no preferential localisation. T cell protrusions were highly enriched in actin, as shown by imaging the actin probe LifeAct-GFP (Fig. 1B) and quantifying its *enrichment* (Fig. 1C), which was significantly higher than the CAAX marker *enrichment*. On the other hand, analysis of confocal imaging of endogenously tagged Lck, LAT and CD45 yielded a *relative enrichment* around 1 (Fig. 1D,E), suggesting no preferential localisation of these proteins to actin-rich protrusions prior to activation. STED imaging further revealed the lack of protein enrichment in specific sub-domains within the protrusions in resting T cells (Fig. 1D).

## Contact with the target cell and clustering of CARs is enhanced by actin protrusions

As resting T cells showed no enrichment of the signalling proteins investigated in actin-rich protrusions, we next examined the localisation and dynamics of signalling proteins during early cell–cell interactions to assess the potential role of protrusions in early signalling. To trigger cell–cell interactions and T-cell signalling, we employed a second-generation HER2-specific CAR containing an ectodomain derived from the HER2-specific FRP5 antibody and a CD28-CD3ζ endodomain (Ahmed et al, 2015). Live-cell STED imaging of HER2-CAR in resting Jurkat T cells highlights pre-clustering of the receptor (Fig. EV3A), however, quantification of the CAR *enrichment* showed no preferential localisation to actin-rich protrusions (Fig. EV3B). To follow the rearrangement of the receptor upon contact with a target cell, we co-cultured Jurkat T cells co-expressing GFP-tagged HER2-CAR and the membrane marker SNAP-CAAX with HER2$^+$ SK-BR-3 cancer cells, which exhibit elevated HER2 receptor levels on their cell surface (Dai et al, 2017) (Fig. 2A; Movie EV1).

Live-cell confocal microscopy allowed us to follow cell–cell interactions from before cell–cell interaction until immunological synapse formation (Fig. 2A; Movie EV1). The first contact detected between the T cell and the target cell usually occurred on actin-rich protrusions (Figs. 2Aii and EV4B). After that, the HER2-CAR quickly clustered at protrusions (Fig. 2Aiii) and more protrusive structures polarised to the synapse (Fig. 2Aiv). These structures eventually collapsed (Fig. 2Av,vi), giving rise to a flattened immunological synapse. We could observe a similar behaviour in human primary CD4$^+$ T cells (Fig. EV5; Movie EV2). In addition, we often observed that projections from the SK-BR-3 cells contacted the Jurkat T cell, and HER2-CARs were subsequently accumulated at those contacts (Fig. EV4A,C; Movie EV3). This suggests that protrusions in general, and not exclusively from the T cell itself, can drive CAR accumulation at the interaction site. To evaluate the contribution of actin-rich protrusions and the main

cell body to CAR clustering during early cell–cell interactions, we segmented these regions using our newly developed image analysis pipeline (Fig. 2B; Movie EV4). This allowed to quantify the involvement of protrusions and main cell body membranes in cell–cell contact over time (Fig. 2C). Our analysis revealed that actin-rich protrusions were the predominant feature at the T cell-target interface during the initial 2 min of interaction. To determine whether different plasma membrane regions—protrusions versus the main body—differ in their capacity to initiate clusters, we calculated the *relative enrichment* of the HER2-CAR in both regions, comparing contacting and non-contacting areas (Fig. 2B). During the first 2 min, all membranes in contact with the target showed enhanced CAR *relative enrichment* (Fig. 2D); however, *relative enrichment* was significantly higher in protrusion contacts in comparison to main body membrane contacts. In addition, quantitative analysis of the CAR *relative enrichment* after the first contact detected (set at $t = 0$ s) revealed that CAR accumulated both more rapidly and to a greater extent in protrusive contacts (Fig. 2E). A significant increase in CAR signal in protrusions was already noticeable 4 s post-contact, while CAR enrichment in the main body membrane occurred only 30 s post-contact. Overall, we demonstrate that protrusions promote the enrichment and clustering of CAR.

## CAR clusters in protrusions are activation hotspots

Taking advantage of our image analysis pipeline, we wanted to investigate whether CAR clusters in protrusions are activation hotspots by determining the dynamics of recruitment of downstream machinery to favour signal initiation. One of the earliest signalling events following CAR engagement and triggering is the recruitment of the tyrosine kinase ZAP-70 to the phosphorylated ITAMs of the cytoplasmic CD3ζ tail of the CAR. Therefore, ZAP-70 recruitment to the membrane is a reliable indicator of proximal CAR signalling (Gudipati et al, 2020). After its recruitment and activation by Lck phosphorylation, ZAP-70 phosphorylates the adaptor LAT (Zhang et al, 1998), that will scaffold a signalling hub for downstream signal propagation via calcium- and MAPK-dependent signalling pathways (Balagopalan et al, 2015). LAT clustering is thus another downstream readout of receptor activation.

To determine whether these CAR clusters are activation hotspots and whether increased CAR accumulation in protrusions correlates to an enhanced ZAP-70 recruitment, we expressed the HER2-CAR-GFP in a Jurkat T-cell line expressing endogenously Halo-tagged ZAP-70 and followed the interaction with the target cell via confocal microscopy (Fig. 3A; Movie EV5). ZAP-70

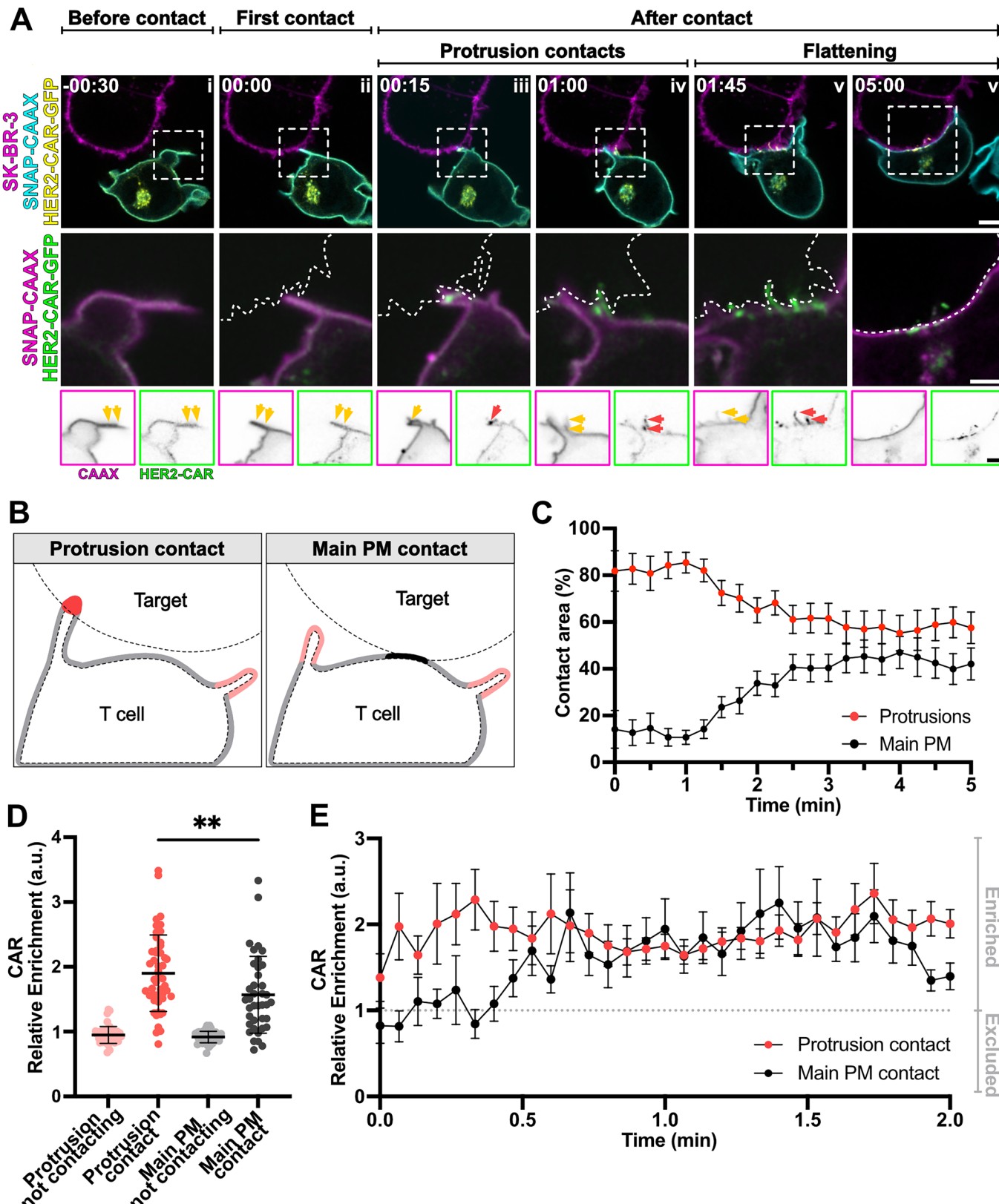

**Figure 2. Contact with the target cell and early clustering of chimeric antigen receptors (CAR) is governed by protrusions.**

(A) Time-lapse confocal imaging of a Jurkat T cell expressing HER2-CAR-GFP, SNAP-CAAX (labelled with BG-JF$_{571}$) and interacting with a SK-BR-3 cell (labelled with CellMaskOrange). Dashed line describes the outline of the SK-BR-3 cell. Yellow arrows highlight protrusions, red arrows highlight the appearance of CAR clusters. (B) Schematic representation of the four different plasma membrane regions segmented using the image analysis pipeline described in the Methods section and Appendix Fig. S1. (C) Percentage of contact area between the SK-BR-3 and the Jurkat T cell mediated by protrusions and by the main body membrane. $n = 20$ independent cell–cell interaction events captured with confocal microscopy at a 15 s/frame rate are plotted. The graph shows mean values and standard error of the mean (s.e.m.) error bars. (D) Mean *relative enrichment* of HER2-CAR-GFP in the four different regions described in (B) during the first 2 min of the interaction with the SK-BR-3. $n = 49$ independent cell–cell interaction events were analysed. *P* value of paired *t* test is 0.0094. The graph shows mean values and s.d. error bars. (E) *Relative enrichment* of the CAR in the protrusions and main body membrane contacts over time. $n = 29$ captured with confocal microscopy at a 4 s/frame rate are plotted. Graph shows mean values, s.e.m. error bars. BG benzylguanine (SNAP-tag substrate). Scale bars, 5 μm (confocal overviews), 2 μm (crops).

recruitment to contacts immediately followed CAR clustering (Fig. 3Aii), after the first contact was detected (Fig. 3Ai). Although contacts via the main body membrane were uncommon, they also led to successful ZAP-70 recruitment. The recruitment of ZAP-70 to T cell/target contacts was assessed by calculating the *membrane-bound* ZAP-70 as the difference between the signal in protrusion/main body membrane contacts and protrusion/main body membrane not contacting the target. *Relative membrane-bound* ZAP-70 was then calculated using the CAAX marker as a reference (Fig. 3B,C). ZAP-70 was recruited faster (Fig. 3B) and in a greater magnitude (Fig. 3C) to contacts mediated through protrusions than through main body membranes. Segmentation of CAR clusters allowed us to analyse ZAP-70 recruitment to engaged CARs. Interestingly, the ratio of the *membrane-bound* ZAP-70 to CAR *enrichment* (Fig. 3D) was significantly higher for CAR clusters in protrusions. This observation suggests that CARs are not only more efficiently accumulated but also more efficiently activated at protrusions.

We then asked whether enhanced ZAP-70 recruitment in protrusions was accompanied by faster and greater LAT clustering. For this, we employed a Jurkat T-cell line expressing LAT[EN]-Halo and HER2-CAR-GFP and followed the first minutes of interaction with the target SK-BR-3 cells (Fig. 4A,B; Movie EV6). Shortly after target cell contact (Fig. 4Aii)—following CAR clustering and ZAP-70 recruitment—LAT rapidly clustered within actin-rich protrusions, typically between 8 and 12 s post-contact (Fig. 4B). In contrast, LAT clusters formation at main body interactions sites was delayed, with enrichment becoming apparent only at $t = 32$ s post-contact (Fig. 4B). Overall, LAT clustering during the first minute of cell–cell interaction was significantly higher in protrusion contacts than in the main body membrane contacts (Fig. 4C), indicating that clustering of LAT is favoured in actin-rich protrusions. Interestingly, live-cell confocal microscopy highlighted clusters forming within cell–cell contacts in protrusions and then segregating from the HER2-CAR-positive cluster (Fig. 4Aii–iv). It is important to note that segregation of LAT clusters from contacts could affect the quantification of LAT *relative enrichment* at protrusions and main body membrane differently, as clusters on the cell body may be able to more easily diffuse away from the imaged plane.

Live-cell STED imaging of cells expressing LAT[EN]-Halo and the membrane marker SNAP-CAAX confirmed the localisation of LAT clusters to the plasma membrane of protrusions and highlighted the segregation of the LAT clusters from the CAR clusters (Fig. 4D,E). Segregation of LAT from the membrane contact could be linked to the internalisation and degradation of LAT (Balagopalan et al, 2007). Altogether, CAR clusters in protrusions are better activation

hotspots than clusters derived from interactions via the main body, as demonstrated by the accelerated kinetics of recruitment of downstream activation machinery.

## While Lck remains uniformly distributed, CD45 is excluded from target contacts

To better understand the mechanism behind the increased clustering of CAR and increased recruitment of signalling proteins in protrusions, we employed our quantitative live-cell imaging pipeline to assess the dynamics of localisation of negative (the phosphatase CD45) and positive (the tyrosine kinase Lck) regulators of CAR activation. According to the kinetic segregation model (Davis and van der Merwe, 2006; Xiao et al, 2022), the physical exclusion of the large phosphatase CD45 from close contacts would allow the CAR to be phosphorylated by the kinase Lck unopposed.

To monitor the dynamics of recruitment of the kinase Lck, we expressed the HER2-CAR in a gene-edited Jurkat T cell line expressing Lck[EN]-Halo (Fig. 5) and performed time-lapse confocal microscopy to assess any changes in the localisation of the protein upon contact with the target cell. Interestingly, we could not observe any macroscale-level rearrangement of Lck on the T-cell membrane, whether in contact with the target cell or not (Fig. 5A; Movie EV7). While CAR clustering (Fig. 5Aiii) was prominent after the first contact was detected (Fig. 5Aii), Lck remained evenly distributed on the PM of T cells. Unbiased quantitative analysis highlights that Lck *enrichment* in various PM regions did not show any significant changes over time and upon cell–cell contact (Fig. 5B). No rearrangement of Lck[EN]-Halo in protrusions during the first minute of the interaction was detected (Fig. 5C). Furthermore, live-cell STED imaging of Lck[EN]-Halo cells expressing the membrane marker SNAP-CAAX could not highlight any nanoscale rearrangement of Lck within protrusions upon target cell contact (Fig. 5D, E). Taken together, this indicates that no macroscale-level rearrangement of Lck supported the heightened CAR activation at protrusions.

We then performed the same experiment with gene-edited Jurkat T cells expressing CD45[EN]-Halo (Fig. 6). Strikingly, as the first T cell-target cell contact was detected (Fig. 6Aii), CD45 became excluded from the CAR cluster formed in protrusions (Fig. 5Aiii; Movie EV8). CD45 signal intensities markedly decreased at the sites of initial protrusion contact and CAR clustering (Fig. 6Aii–iv, line profiles). When analysing the *relative enrichment* of CD45 over time (Fig. 6B), we could observe that the exclusion of the protein in contacts with the target cell was immediate ($t = 0$ s). On the contrary, the decrease of CD45 signal at the main body membrane

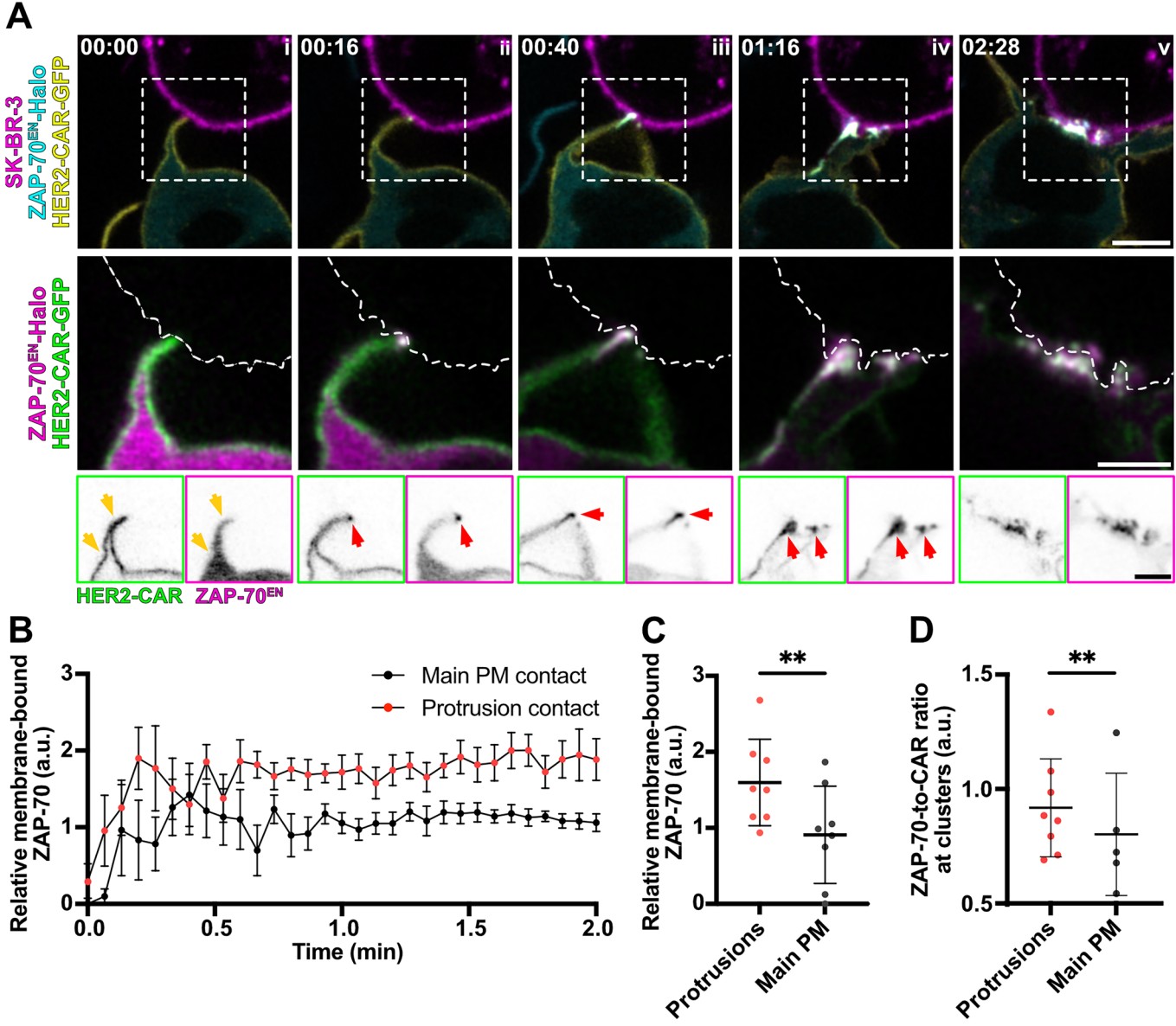

**Figure 3. CAR clusters in protrusions are activation hotspots.**

(A) Time-lapse confocal imaging of Jurkat T cells expressing HER2-CAR-GFP, ZAP-70EN-Halo labelled with CA-JFX$_{650}$ interacting with a SK-BR-3 cell stained with CellMaskOrange. Dashed line describes the outline of the SK-BR-3 cell. Yellow arrows point at protrusions, red arrows highlight ZAP-70 and CAR clusters, localised to protrusions. (B) *Relative membrane-bound* of ZAP-70EN-Halo in the protrusions and main body membrane contacts over time. $n = 8$ independent cell–cell interaction events captured with confocal microscopy at a 4 s/frame rate are plotted. The graph shows mean values, s.e.m. error bars. (C) Mean *relative membrane-bound* of ZAP-70EN-Halo in protrusions and the main body membrane within the first minute of interaction with the target cell. $n = 8$ independent cell–cell interaction events were analysed. The graph shows mean values, s.d. error bars. *P* value of paired *t* test is 0.0010. (D) Mean ratio of *relative membrane-bound* ZAP-70EN-Halo to HER2-CAR-GFP *enrichment* in protrusions and main body membrane within the first minute of interaction with the target cell. $n = 8$ independent cell–cell interaction events were analysed. The graph shows mean values and s.d. error bars. *P* value of paired *t* test is 0.0019. CA chloroalkane (Halo tag substrate). Scale bars, 5 µm (confocal overviews), 2 µm (crops).

contacts only became apparent at later stages of cell–cell interaction ($t = 40$ s). Accordingly, the average *relative enrichment* of CD45 at protrusion contacts was significantly lower than that at main body membrane contacts (Fig. 6C). Furthermore, the CD45-per-CAR ratio at the segmented CAR clusters was significantly lower for those clusters located at T-cell protrusions (Fig. 6D). While CD45 exclusion at protrusion-mediated contacts was only modestly apparent by confocal microscopy, live-cell super-resolution STED

imaging provided a more detailed view—resolving individual protrusions and membrane contours—clearly highlighting CD45 exclusion at sites of CAR clustering (Fig. 6E, F). Altogether, this data indicates that CD45 exclusion might be faster and more efficient at contacts created through protrusive structures, explaining the increased clustering and activation of CARs in protrusions.

Next, we wanted to test whether exclusion of CD45 depends on CAR signalling or is simply driven by the physical interaction

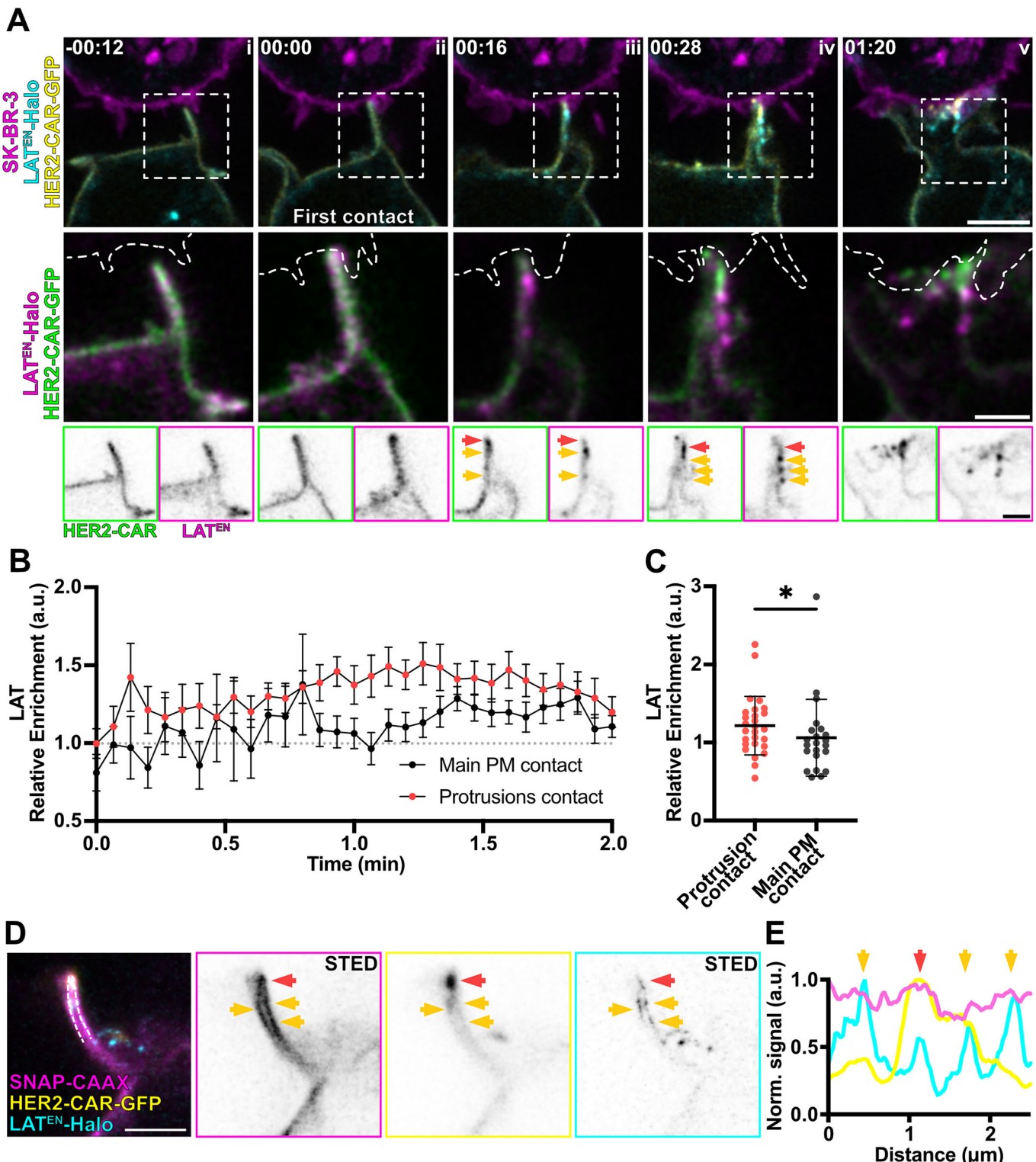

**Figure A:** SK-BR-3 / LAT<sup>EN</sup>-Halo / HER2-CAR-GFP time series: -00:12 (i), 00:00 (ii, First contact), 00:16 (iii), 00:28 (iv), 01:20 (v). Middle row: LAT<sup>EN</sup>-Halo / HER2-CAR-GFP. Bottom row labeled HER2-CAR and LAT<sup>EN</sup>.

**B:** LAT Relative Enrichment (a.u.) vs Time (min); Main PM contact, Protrusions contact.

**C:** LAT Relative Enrichment (a.u.); Protrusion contact vs Main PM contact; *.

**D:** SNAP-CAAX (magenta), HER2-CAR-GFP (yellow), LAT<sup>EN</sup>-Halo (cyan); STED.

**E:** Norm. signal (a.u.) vs Distance (μm).

between CAR and antigen. The indication that this may be solely a physical process came from the observation that CAR *relative enrichment* increased around $t = 4\,s$ (Fig. 2Aiii,C), while CD45 exclusion was evident upon first contact detected at $t = 0\,s$ (Fig. 6Aii,B). To assess whether the process is independent of CAR signalling, we expressed a truncated version of the CAR in the

CD45<sup>EN</sup>-Halo Jurkat T cell line (Fig. 7). The truncated CAR only harbours the extracellular HER2-binding domain, a transmembrane domain and a GFP tag and is thus unable to signal. We assessed the rearrangement of the truncated CAR and CD45 in time-lapse experiments upon contact with the target cell (Fig. 7A; Movie EV9). We found that CD45 exclusion at contact sites

◀ **Figure 4.  LAT clusters form at protrusion contacts.**

(A) Time-lapse confocal imaging of Jurkat T cells expressing HER2-CAR-GFP, LAT$^{EN}$-Halo labelled with CA-JFX$_{650}$ interacting with a SK-BR-3 cell stained with CellMaskOrange. Dashed line describes the outline of the SK-BR-3 cell. Arrows highlight the formation of LAT$^{EN}$ clusters in actin-rich protrusions and segregation from CAR clusters. Yellow arrows point at a LAT cluster colocalising with a HER2-CAR cluster, and red arrows point at a LAT cluster segregated from a CAR cluster. (B) *Relative enrichment* of LAT in actin-rich protrusions contacts and main body membrane contacts over time. $n = 22$ independent cell–cell interaction events are plotted. Graph shows mean values, s.e.m. error bars. (C) Mean *relative enrichment* of LAT in protrusions and main body membrane within the first minute of interaction with the target cell. $n = 22$ independent cell–cell interaction events were analysed. Graph shows mean values, s.d. error bars. P value of paired $t$ test is 0.0367. (D) Live-cell confocal and STED microscopy of HER2-CAR-GFP (confocal), SNAP-CAAX (STED) and LAT$^{EN}$-Halo (STED) labelled with BG-JF$_{571}$ and CA-JFX$_{650}$ of the protrusion of a Jurkat T cell interacting with an unstained SK-BR-3. Yellow arrows point at a LAT cluster colocalising with a HER2-CAR cluster, and red arrows point at a LAT cluster segregated from a CAR cluster. (E) Line profile of the 3-colour image shown in (D). Arrows in the graph correspond to the areas highlighted by the same arrows in (D). CA chloroalkane (HaloTag substrate), BG benzylguanine (SNAP-tag substrate). Scale bars, 5 µm (confocal overviews), 2 µm (crops), 1 µm (STED).

happened instantaneously (Fig. 7Aii, $t = 0$ s), followed by a clear clustering of the truncated CAR (Fig. 7Aiii, $t = 16$ s). This observation mirrors that seen with CARs with an intact signalling domain. However, despite the formation of initial contacts, the membrane of most cells expressing the truncated CAR did not flatten to form a mature synapse (Fig. 7Av), indicating incomplete activation. Further, comparison of the *relative enrichment* of CD45 at protrusion contacts stabilised by the intact and truncated CAR, did not show any significant difference in CD45 exclusion (Fig. 7B). Altogether, these results indicate that CD45 exclusion at cell–cell contacts is CAR signalling-independent and precedes CAR clustering and recruitment of downstream machinery.

## Discussion

In this study, we establish a quantitative, dynamic framework that captures macromolecular rearrangements in relation to membrane topography on the surface of T cells—from before cell–cell contact to immunological synapse formation—all within living, unperturbed cells. This approach offers unprecedented insight into the spatial and temporal coordination of signalling events in their native context. Our findings indicate that prior to activation, key signalling machinery lacks any enrichment in actin-rich protrusions (Fig. 8A,B). Here, we define actin-rich protrusions as plasma membrane extensions enriched in F-actin (Fig. 1B,C), which enabled us to assess their general function. Upon contact with the target cell, we observed a marked increase in CAR and LAT clustering, enhanced ZAP-70 recruitment and CD45 exclusion within protrusions compared to the main cell body membrane regions (Fig. 8C vs Fig. 8D). Overall, this study reveals that actin-rich protrusions are privileged sites for the initiation of CAR-mediated T-cell activation.

Our observation that HER2-CAR, Lck, LAT and CD45 do not exhibit preferential localisation to protrusions (Figs. 1E, EV3B, and 7B) differs from a previous report suggesting that Lck and LAT may be pre-assembled in protrusive structures known as "microvilli" (Ghosh et al, 2020). Similar discrepancies have been noted in studies on TCR localisation, likely due to differences in imaging methodologies. For example, while some studies reported TCR enrichment in protrusions (Ghosh et al, 2020; Jung et al, 2016) others found no such enrichment (Cai et al, 2022). To address these discrepancies, we sought to characterise the resting distribution of signalling proteins in living cells using a purely optical super-resolution microscopy technique such as STED (Lukinavičius et al, 2024). By doing so, we aimed to distinguish between pre-existing

spatial arrangements and dynamic protein redistributions that occur specifically upon target engagement. Our findings strongly suggest that actin-rich protrusions do not aid CAR signalling through pre-enrichment of signalling molecules and highlight the importance of live-cell imaging for accurately capturing the native organisation of signalling components.

Upon contact with the target cell, actin-rich protrusions became the primary structure mediating early interaction at the interface (Fig. 2C). Even after a partial flattening of the T cell, we could observe some protrusive interdigitations with the breast cancer cell (Fig. 2Aiv), which have also been previously reported (Sanderson and Glauert, 1979). By imaging the HER2-CAR rearrangement in a Jurkat T cell interacting with the target, we could monitor how the different membrane regions contribute to close contact formation and receptor recruitment. The protrusive behaviour of Jurkat T cells is identical to that of peripheral blood-derived CD4+ primary T cells, suggesting analogous mechanisms (Fig. EV5). We found that, although CARs accumulated at both protrusions and the main body membrane interacting with the target, CAR *enrichment* was consistently higher in protrusions (Fig. 2D). This suggests that protrusions are more efficient at initiating close contacts. In some cases, we could observe how target cell protrusions engaged the T cell main body membrane and triggered CAR clustering (Fig. EV4A). This observation aligns with previous studies showing that dendritic F-actin structures facilitate T-cell signalling and priming (Leithner et al, 2021). Since T-cell protrusions exert pushing forces that promote close apposition of membranes (Sage et al, 2012; Sanderson and Glauert, 1979), it is plausible that protrusions from the target cell may be able to do the same.

The increased HER2-CAR clustering in actin-rich protrusions is accompanied by an enhanced ZAP-70 accumulation at protrusions (Fig. 3B,C), indicating that these clusters function as activation hotspots from the moment of their formation. The ZAP-70-per-CAR ratio is higher in CAR clusters localised at protrusions in comparison to CAR clusters on the main body membrane (Fig. 3D), suggesting CAR activation is enhanced at protrusions. ZAP-70 recruitment is followed by faster (Fig. 4B) and more pronounced (Fig. 4C) LAT clustering at the same sites. Although this result could be simply explained by the increased HER2-CAR/ZAP-70 signal at protrusions, the architecture of the protrusion itself might be relevant for enhancing the immune response, as we speculate that negative curvature may favour accumulation of signalling proteins by reducing diffusion within membranes. In addition, we could observe LAT cluster segregation from protrusion contacts. It is possible that such events could also occur at clusters on the main body membrane and may have been missed because of our 2D imaging modality. Our data suggest that actin-

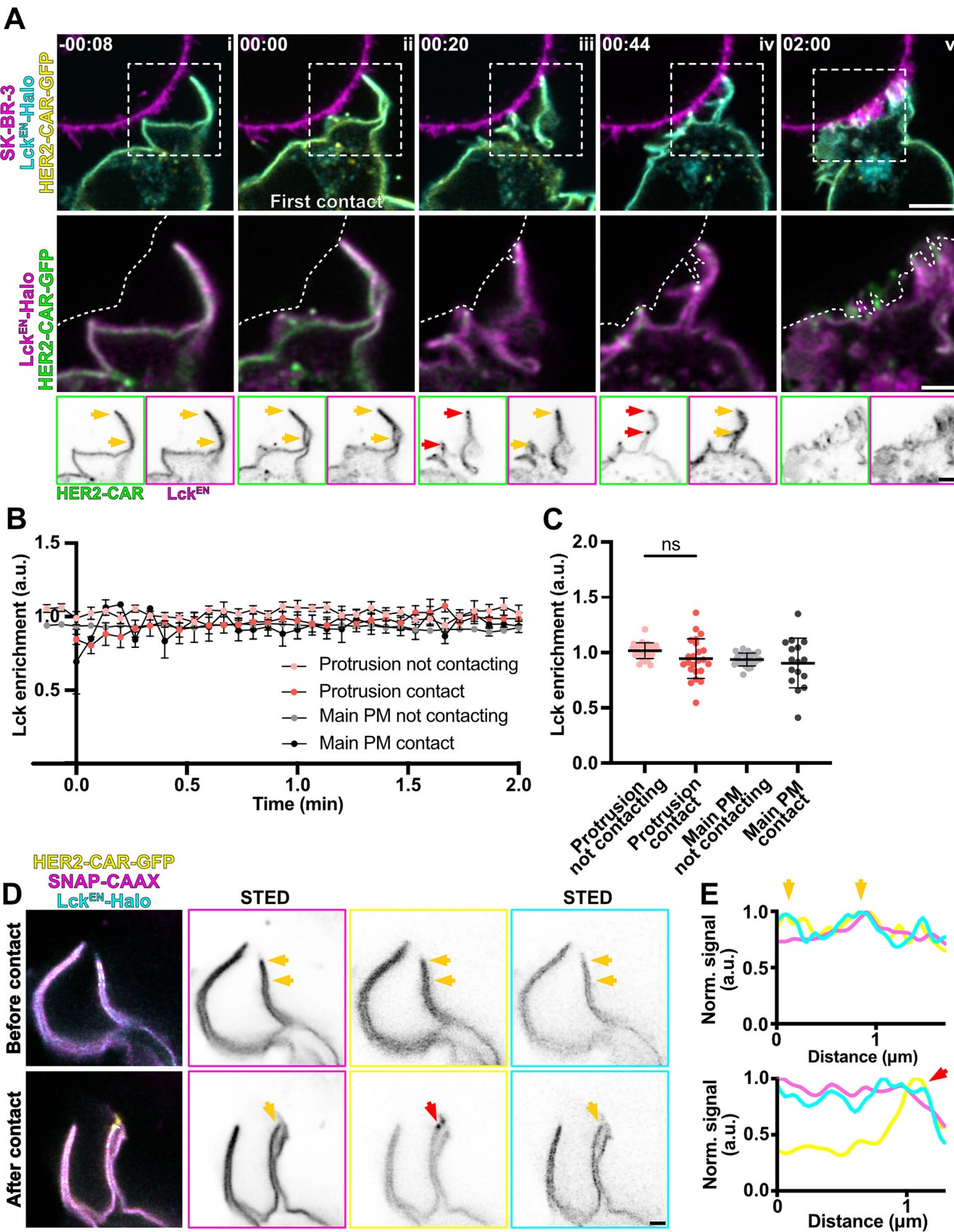

**Figure 5. Lck remains evenly distributed on the plasma membrane of T cells upon CAR engagement.**

(A) Time-lapse confocal imaging of Jurkat T cells expressing HER2-CAR-GFP, Lck$^{EN}$-Halo labelled with CA-JFX$_{650}$ interacting with a SK-BR-3 cell stained with CellMaskOrange. The dashed line describes the outline of the SK-BR-3 cell. Yellow arrows highlight protrusions and the lack of rearrangement of Lck at the macroscale, while red arrows point at CAR clusters at protrusions upon contact with the target cell. (B) *Enrichment* of Lck in the four different membrane regions segmented as described in Fig. 2B. $n = 26$ independent cell–cell interaction events are plotted. The graph shows mean values and s.e.m. error bars. (C) Mean *enrichment* of Lck$^{EN}$ in protrusion and main body membrane, contacting or not contacting the target during the first minute of the interaction. $n = 23$ independent cell–cell interaction events were analysed. The graph shows mean values and s.d. error bars. *P* value of paired *t* test is 0.0939 for protrusion regions and 0.4764 for body membrane regions. (D) Live-cell confocal and STED microscopy of HER2-CAR-GFP (confocal), SNAP-CAAX (STED) and Lck$^{EN}$-Halo (STED) labelled with BG-JF$_{571}$ and CA-JFX$_{650}$ of a protrusion of a Jurkat T cell before and after interacting with an unstained SK-BR-3. The sections corresponding to the line profile (E) is shown in the three-colour image. Yellow arrows highlight protrusions and the lack of rearrangement of Lck at the macroscale, while the red arrow points at the CAR cluster in a protrusion upon contact with the target cell. CA chloroalkane (HaloTag substrate), BG benzylguanine (SNAP-tag substrate). Scale bars, 5 μm (confocal overviews), 2 μm (crops), 1 μm (STED).

rich protrusions may enhance CAR accumulation (Fig. 2C,D) and activation (Fig. 3D) by excluding CD45 from the interaction site more rapidly (Fig. 6B) and to a greater extent (Fig. 6C,D) than the main body membrane. CD45's large extracellular domain (Chang et al, 2016) exceeds the HER2-CAR/HER2 span, and its size-dependent exclusion from close contacts enables receptor triggering (Davis and van der Merwe, 2006). We have shown that early exclusion of CD45 is CAR signalling-independent (Fig. 7) and precedes CAR recruitment to the contact (Fig. 2D vs Fig. 6C). As the degree of exclusion of CD45 has been related to the "tightness" of the close contacts created by T cells interacting with antibody-coated surfaces (Razvag et al, 2019), our results suggest that actin-rich protrusions create "tighter" contacts, as indicated by a lower CD45 *relative enrichment* and CD45-per-CAR ratio (Fig. 6C,D). The more efficient CD45 exclusion at protrusions was sufficient to provide a nanoscale environment that enhanced CAR activation even in the absence of macroscale rearrangement of the positive regulator of activation, Lck (Fig. 5). This suggests that CARs are activated through kinetic segregation of CD45, in agreement with a recent study reporting that CAR activation is dependent on the length of the CAR/ligand complex (Xiao et al, 2022).

Previous work shows that protrusions are hyper-stabilised during CAR-mediated activation (Beppler et al, 2023) and that this stabilisation depends on ligand density and receptor affinity. Hyper-stabilised signalling foci (as also observed here) could cause multi-focal synapse characteristics of some CARs (Beppler et al, 2023; Davenport et al, 2018; Gudipati et al, 2020; Xiong et al, 2018). In a physiological T cell-APC contact, where pMHC is less dense, and the affinity of the interaction is several orders of magnitude lower, protrusions are likely shorter-lived. Under these conditions, TCR accumulation at protrusions may be considerably less pronounced than what we observe here for the HER2-CAR. As CD45 exclusion is signalling-independent (Fig. 7), we speculate that there may be no differences in exclusion dynamics between CAR- and TCR-dependent activation. In addition, the lack of measurable Lck rearrangement in the CAR system (Fig. 5) is in agreement with reports of reduced Lck recruitment in mature CAR synapses in comparison to the TCR (Davenport et al, 2018). This could possibly be explained by the absence of CD4-bound Lck brought in proximity of the CAR by MHC-CD4 interactions (Artyomov et al, 2010) or a lower reliance of CD28-based CARs on Lck phosphorylation (Wu et al, 2023).

In summary, combining live-cell super-resolution imaging of endogenously tagged proteins with quantitative analysis of membrane topography and protein dynamics allowed us to understand the role of membrane topography during cell–cell

interactions. This study focuses on understanding how actin-rich protrusions drive early cell–cell interactions and signalling in CAR-mediated activation. The established cell lines and methodologies could be employed to investigate fundamental questions about the role of membrane topography in TCR-mediated activation and its function at later timepoints of interaction with a target.

## Methods

### Reagents and tools table

| Reagent/resource | Reference or source | Identifier or catalogue number |
|---|---|---|
| **Experimental models** | | |
| Jurkat T cells | Leibniz Institute DSMZ – German Collection of Microorganisms and Cell Cultures GmbH | ACC 282 |
| Normal Human Peripheral Blood T-Cells, CD4+ | BioCat GmbH | SER-CD4-PLUS-T-F-ZB |
| SK-BR-3 | Leibniz Institute DSMZ – German Collection of Microorganisms and Cell Cultures GmbH | ACC 736 |
| **Recombinant DNA** | | |
| pH6HTC His6HaloTag® T7 | Promega | JN874647 |
| pSNAPf | Addgene | #58186 |
| pEGFP-N1 | Addgene | #6085-1 |
| GFP-CAAX | Addgene | #86056 |
| SpCas9 pX330 | Addgene | #42230 |
| ARF3-LAP-Halo-2xALFA-LoxP-G418$^r$-LoxP HR plasmid | Wong-Dilworth et al, 2023 | |
| ARF1-Halo homology repair plasmid | Bottanelli et al, 2017 | |
| AP1μA$^{EN}$-Halo-ALFA-polyAG418 plasmid | Stockhammer et al, 2024 | |
| **Antibodies** | | |
| Anti-Lck | Santa Cruz | sc-433 |
| Anti-ZAP-70 | Santa Cruz | sc-32760 |
| Anti-LAT | Santa Cruz | sc-53550 |
| Anti-CD45 | ProteinTech | 20103-1-AP |

| Reagent/resource | Reference or source | Identifier or catalogue number |
|---|---|---|
| Anti-HaloTag | Promega | G921A |
| Anti-β-Actin | Cell Signaling | 8H10D10 |
| Anti-p-Tyr-100 | Cell Signaling | 3584 s |
| Anti-phospho-LAT | Cell Signalling | Tyr191 |
| Anti-mouse IgG HRP | abcam | AB6789 |
| Anti-rabbit IgG HRP | abcam | AB6721 |
| Anti-CD3, OKT3 clone | Invitrogen | 16-0037-85 |
| Anti-CD28 | Biolegend | 302933 |
| **Oligonucleotides and other sequence-based reagents** | | |
| Primers used for cloning | This study | Table EV1 |
| gBlocks used for cloning | This study | Table EV2 |
| Guide sequences for gene modification with CRISPR/Cas9 | This study | Table EV3 |
| **Chemicals, enzymes and other reagents** | | |
| **Dyes** | | |
| CellMask Orange Plasma Membrane Stain | Plasma membrane | Invitrogen |
| JF$_{571}$-BG | SNAP Tag | Lavis lab |
| JFX$_{650}$-BG | SNAP Tag | Lavis lab |
| JF$_{650}$-Chloroalkane (CA) | Halo Tag | Lavis lab |
| **Enzymes** | | |
| BamHI-HF | New England Biolabs (NEB) | R3131S |
| Bpil | Thermo Fisher | FD1014 |
| EcoRI-HF | NEB | R3101S |
| HindIII-HF | NEB | R3104S |
| Nhe-HF | NEB | R3131S |
| NotI | NEB | R3189S |
| Pfu polymerase | Roboklon | E1114 |
| Quick CIP phosphatase | NEB | M0525S |
| T4 ligase | NEB | M0202S |
| **Chemicals** | | |
| 2-Propanol (Isopropanol) | Roth | 6752.4 |
| 20x Bolt MES SDS Running Buffer | Thermo Fisher | B000202 |
| β-Mercaptoethanol | Roth | 4227.1 |
| Bromophenol | VWR Chemicals | 87795.180 |
| Bovine serum albumin (BSA) | Roth | 8076.4 |
| Complete EDTA-free Protease Inhibitor | Sigma-Aldrich | 4693132001 |
| deoxynucleotide triphosphates (NTPs) | NEB | N0447S |
| Dulbecco's Phosphate Buffered Saline (D-PBS), 10x | Corning | 20-031-CV |
| EDTA | Biozym Scientific | 831234 |

| Reagent/resource | Reference or source | Identifier or catalogue number |
|---|---|---|
| EGTA | abcr | AB304133 |
| Ethanol | Roth | 5054.4 |
| Foetal Bovine Serum (FBS) | Corning | 35-079-CV |
| Geneticin disulfate (G418) | Roth | CP11.3 |
| Gibson Assembly Master Mix | NEB | E2611 |
| Glycerin | Roth | 3783.1 |
| GlutaMAX | Gibco | 35050061 |
| 4-(2-hydroxyethyl)-1-piperazineethanesulfonic acid (HEPES) | Roth | 9105.2 |
| Human Recombinant IL-2 | Stemcell Technologies | 78036 |
| ImmunoCult-XF T cell expansion medium | Stemcell Technologies | 10981 |
| ImmunoCult CD3/CD28 T cell activator | Stemcell | 10971 |
| Milk powder | Roth | T145.3 |
| NP40 (IGEPAL® CA-630 for molecular biology) | Sigma-Aldrich | MKCG8897 |
| Opti-MEM I Reduced Serum Medium, GlutaMAX Supplement | Gibco | 51985026 |
| Penicillin-Streptomycin | Corning | 30-002-CI |
| Pfu polymerase | Roboklon | E1114 |
| Phosphate Buffered Saline (1x) | Gibco | 10010-015 |
| Roswell Park Memorial Institute (RPMI) 1640 | Gibco | 21875034 |
| Rotiphorese 10x TAE buffer | Roth | T845.2 |
| Sodium dodecyl sulfate (SDS) | Roth | 1057.1 |
| Sodium azide (N$_3$Na) | Roth | 4221.1 |
| Sodium deoxycholate | Sigma-Aldrich | D6750 |
| Sodium orthovanadate (Na$_3$VO$_4$) | Sigma-Aldrich | 450243-5OG |
| Sulphuric acid | Roth | 9896.2 |
| T4 ligase buffer | NEB | B0202S |
| Tris-HCl | Roth | T845.2 |
| Triton-X 100 | Sigma-Aldrich | T8787 |
| Tween20 | Roth | 9127.1 |
| **Software** | | |
| Affinity Designer | Serif Europe | |
| Benchling | https://benchling.com | |
| CellProfiler | Carpenter et al, 2006 | |
| Grammarly Pro | Superhuman | |
| Graphpad Prism 9 | Graphpad Software | |
| ImageJ (Fiji) | Schindelin et al, 2012 | |
| Imspector v16.3.16118 | Abberior | |

| Reagent/resource | Reference or source | Identifier or catalogue number |
|---|---|---|
| Office Excel | Microsoft | |
| **Other** | | |
| 8-well chambered cover glass | IBL | C8-1.5-H-N 220.140.082 |
| Bolt™ 4–12%, Bis-Tris, 1.0 mm, Mini-protein-gels | Invitrogen | NW04122BOX |
| Cytiva Amersham Protran Supported NC Nitrocellulose Membrane | Fisher Scientific GmbH | 15209814 |
| Electroporation cuvettes | Fisher Scientific GmbH | 15542423 |
| ELISA MAX™ Deluxe Set Human IL-2 kit | Biolegend | 431815 |
| Nunc™ MaxiScorp™ ELISA plates, Uncoated | Biolegend | 431815 |

## Mammalian cell culture

Jurkat T cells (Cat # ACC 282 DSMZ) and SK-BR-3 cells (Cat # ACC 736 DSMZ) were grown in a humidified incubator at 37 °C with 5% $CO_2$ in Roswell Park Memorial Institute medium (RPMI, Gibco) supplemented with 10% foetal bovine serum (Corning), 100 U/L penicillin and 0.1 g/L streptomycin (FisherScientific). Normal human peripheral blood CD4+ primary T cells (Cat # SER-CD4-PLUS-T-F-ZB, BioCat) were grown in ImmunoCult-XF T cell expansion medium (Stemcell Technologies) supplemented with 10% foetal bovine serum, 100 U/L penicillin, 0.1 g/L streptomycin and 10 ng/mL human Recombinant IL-2 (Stemcell Technologies) at 37 °C with 5% $CO_2$.

For transient transfection of plasmids encoding for SNAP-CAAX, HER2-CAR-GFP, HER2-CAR-Halo or truncated HER2-CAR-GFP into Jurkat T cells, a NEPA21 electroporation system was used (Nepa Gene). 5 million cells were washed twice with Opti-MEM (Gibco), resuspended in 90 μl Opti-MEM and mixed with 10 μg of DNA in an electroporation cuvette with a 2-mm gap (Fisher Scientific). The electroporation reaction consists of two poring pulses (150 V, 5 ms length, 50 ms interval, with decay rate of 10% and + polarity) and five consecutive transfer pulses (20 V, 50 ms length, 50 ms interval, with a decay rate of 40% and ± polarity). For transient transfection of SNAP-CAAX and HER2-CAR-GFP in primary T cells, CD4 + T cells were activated with ImmunoCult Human CD3/CD28 T cell Activator (Stemcell Technologies) following the manufacturer's instructions. After 48 h, 3 million cells were electroporated using the NEPA21 system. The electroporation reaction consists of two poring pulses (175 V, 5 ms length, 50 ms interval, with decay rate of 10% and + polarity) and five consecutive transfer pulses (20 V, 50 ms length, 50 ms interval, with a decay rate of 40% and ± polarity).

## Generation of plasmids for overexpression and gene editing

For all plasmids harbouring a Halo-encoding sequence, the HaloTag sequence was taken from pH6HTC His6HaloTag® T7 (Promega, JN874647). For all plasmids harbouring a SNAP-encoding sequence, the SNAP-tag sequence was taken from the pSNAPf (Addgene plasmid #58186). For all plasmids harbouring a GFP-encoding sequence, the GFP sequence was taken from pEGFP-N1 (Addgene plasmid #6085-1). All plasmids were verified through sequencing.

### Cloning of overexpression plasmids

To generate the SNAP-CAAX encoding plasmid, the GFP tag was replaced with a SNAP tag in the GFP-CAAX plasmid (Addgene plasmid #86056). GFP-CAAX was digested with NheI and EcoRI, and the SNAP sequence was amplified from pSNAPf plasmid and ligated into the digested plasmid as an NheI-EcoRI fragment. Sequences of all primers used for fragment amplification are provided in Table EV1.

For the generation of the HER2-CAR-GFP, the truncated HER2-CAR-GFP and the LifeAct-GFP plasmids, a similar strategy was used. A double-stranded DNA fragment with the desired sequence was synthesised (IDT) and cloned into the EcoRI/BamHI linearised eGFP-N1 vector (Clontech) via Gibson's assembly. A detailed list of all gBlocks used for cloning is provided in Table EV2.

For the generation of the HER2-CAR-Halo plasmid, the HER2-CAR-GFP plasmid was digested with BamHI/NotI. A BamHI/NotI Halo fragment was then ligated into the linearised vector.

### Cloning of guide RNA and homology repair (HR) plasmids

Designing of plasmids and cloning were carried out as explained in detail in (Adarska et al, 2025). All guide RNAs were designed using Benchling (https://www.benchling.com). Single-strand sense and anti-sense oligos containing the guide sequence and harbouring BpiI sites were synthesised and cloned via oligo annealing into the SpCas9 pX330 plasmid (addgene plasmid #42230) (Cong et al, 2013), which was previously linearised with BpiI (Life Technologies). A detailed list of guide RNA sequences is provided in Table EV3.

To prevent re-cutting from the Cas9, the protospacer-adjacent motif (PAM) site in the HR plasmids was mutated.

HR plasmid CD45/ZAP-70[EN]-LAP-Halo-polyA-G418[r]. The HR plasmid consists of ~600 bp homology arms in a High Copy Amp plasmid (pMB1) backbone (Twist Bioscience). A glycine-serine linker and BamHI/EcoRI sites were added between the two homology arms for cloning of tags and a resistance cassette for antibiotic selection of positively edited cells. The coding sequences of a localisation and affinity purification (LAP) linker (Cheeseman and Desai, 2005), the HaloTag and a small 2xALFA epitope-tag were obtained from the ARF3-LAP-Halo-2xALFA-LoxP-G418[r]-LoxP HR plasmid (Wong-Dilworth et al, 2023) as a NheI/EcoRI fragment. The polyA sequence and the G418 resistance cassette were obtained via PCR amplification from the ARF1-Halo homology repair plasmid (Bottanelli et al, 2017).

HR plasmid Lck [EN]-Halo-polyA-G418[r]. The HR plasmid consists of ~1000 bp homology arms in a High Copy Amp plasmid (pMB1) backbone (Twist Bioscience). BamHI and EcoRI sites were added for insertion of the Halo tag sequence and the resistance cassette. A short glycine-serine linker was added between two BamHI sites. The HaloTag sequence was obtained from the AP1μA[EN]-Halo-ALFA-polyAG418 plasmid (Stockhammer et al, 2024) as a BamHI/EcoRI fragment.

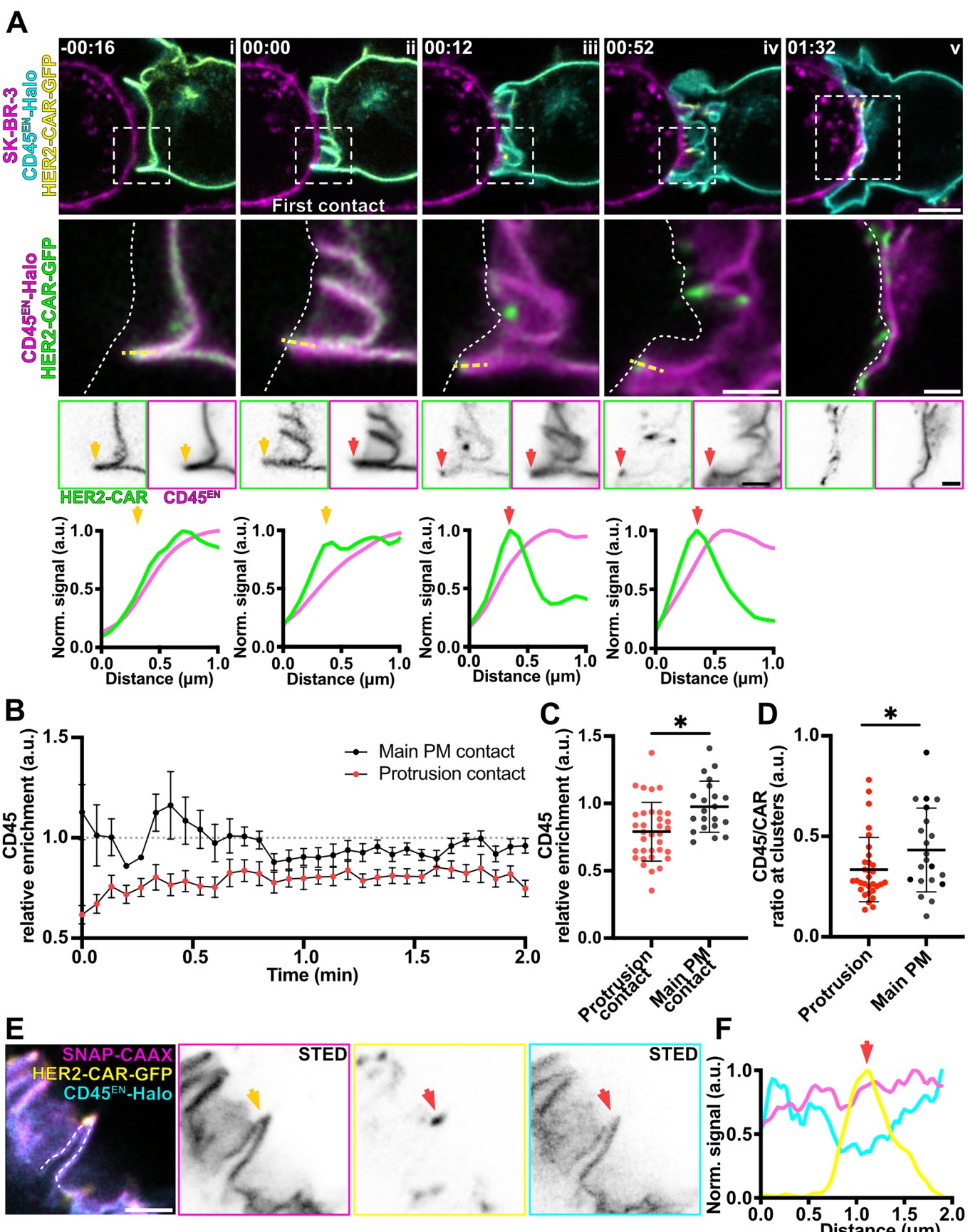

◄ **Figure 6.  CD45 is excluded from contacts with the target cell, and the exclusion is more efficient in actin-rich protrusions.**

(A) Time-lapse confocal imaging of Jurkat T cells expressing HER2-CAR-GFP and CD45[EN]-Halo (labelled with CA-JFX$_{650}$) interacting with a SK-BR-3 cell stained with CellMaskOrange. The dashed line describes the outline of the SK-BR-3 cell. Yellow arrows highlight a protrusion displaying non-preferential localisation of CD45 or the CAR prior to contact with the target cell, and red arrows highlight a membrane domain with decreased CD45 signal and CAR enrichment. Box profiles show HER2-CAR and CD45 signal along the dashed yellow line. (B) *Relative enrichment* of CD45 in actin-rich protrusions contacts and main body membrane contacts over time. $n = 34$ independent cell–cell interaction events are plotted. The graph shows mean values, s.e.m. error bars. (C) Mean *relative enrichment* of CD45 in protrusions and main body membrane within the first minute of interaction with the target cell. $n = 34$ independent cell–cell interaction events were analysed. The graph shows mean values and s.d. error bars. *P* value of paired *t* test is 0.0169. (D) CD45/CAR *enrichment* ratio at CAR clusters localised to protrusions and main body membrane within the first minute of interaction with the target cell. $n = 34$ independent cell–cell interaction events were analysed. The graph shows mean values, s.d. error bars. *P* value of paired *t* test is 0.0175. (E) Live-cell confocal and STED microscopy of HER2-CAR-GFP (confocal), SNAP-CAAX (STED) and CD45[EN]-Halo labelled with BG-JF$_{571}$ and CA-JFX$_{650}$ in a Jurkat T cell interacting with an unstained SK-BR-3. Red arrow points at a HER2-CAR cluster formed at a cell–cell contact to highlight the dramatic decrease of CD45 signal intensity. The yellow arrow shows the position of this CAR cluster and CD45 exclusion region at a protrusion with the SNAP-CAAX membrane reference. (F) Line profile of the 3-colour image shown in (E). Arrow corresponds to the area highlighted in (E). CA chloroalkane (HaloTag substrate), BG benzylguanine (SNAP-tag substrate). Scale bars, 5 µm (confocal overviews), 2 µm (crops), 1 µm (STED).

HR plasmid LAT $^{EN}$-Halo-polyA-G418$^r$.   The HR plasmid consists of ~1000 bp homology arms in a High Copy Amp plasmid (pMB1) backbone from Twist Bioscience. A glycine-serine linker and a NheI and EcoRI sites were added between the two homology arms for insertion of the Halo tag sequence and resistance cassette. The HaloTag, a small 2xHA epitope tag, a polyA and the G418 resistance cassette sequences were amplified from the ARF1-Halo HR plasmid (Bottanelli et al, 2017).

## Generation of CRISPR Knock-In cell lines

For the generation of Jurkat Knock-In cell lines, 4 million cells were washed twice with Opti-MEM (Gibco), resuspended in 90 µl Opti-MEM and mixed with 5 µg of HR plasmid and 5 µg of sgRNA/Cas9 plasmid in an electroporation cuvette with a 2-mm gap. Electroporation was performed as described in "Mammalian cell culture". G418 was added to the cells 3 days after transfection at a concentration of 3 mg/mL (G418), and the media was exchanged every 2–3 days until selection was complete (approximately after 7 days). 0.5% brightest cells for the endogenously tagged Lck, LAT or CD45 cell lines and 0.01% brightest for the ZAP-70 cell line were isolated via Fluorescence-Activated Cell Sorting (FACS).

## SDS page and western blot analysis

For preparation of the samples used in Fig. EV1, 2 million cells were lysed in 200 µl of RIPA buffer [150 nM NaCl (Roth), 0.1% Triton X-100 (Sigma-Aldrich), 0.5% sodium deoxycholate (Sigma-Aldrich), 0.1% SDS (CarlRoth) and 50 mM Tris-Hcl (CarlRoth) pH 8.0, 1× protease inhibitor cocktail (Roche)]. For preparation of the samples used in Fig. EV2, cells were starved overnight and (when required) activated for 5 min in 24-well plates (Corning) coated with anti-CD3 (Invitrogen) and anti-CD28 (Biolegend). Two million cells were lysed in 200 µl of RIPA buffer [150 nM NaCl (Roth), 0.1% Triton X-100 (Sigma-Aldrich), 0.5% sodium deoxycholate (Sigma-Aldrich), 0.1% SDS (Roth) and 50 mM Tris-Hcl (Roth) pH 8.0] supplemented with 1× protease inhibitor cocktail (Roche), 1 mM phenylmethylsulfonyl fluoride (PMSF) and 2 mM Na$_3$VO$_4$.

Cell lysates were clarified by centrifugation for 10 min at 4 °C and 14,000 g and mixed 1:1 with 2× Laemmli buffer [4% SDS, 20% Glycerol (CarlRoth), 120 nM TRIS-Hcl pH 6.8, 0.02%, Bromophenol (VWR Chemicals) and 5% β-mercaptoethanol (CarlRoth)]. Cell

lysates were then loaded on 4–12% SDS-polyacrylamide gels (Life Technologies). After electrophoresis, proteins were transferred to a nitrocellulose membrane (Amersham, Fisher Scientific GmbH) via wet blotting. Membranes were blocked with 5% milk powder (Roth) and 1% BSA (Roth) in PBST [1× PBS (phosphate buffer saline, Corning), 0.5% Tween20 (Roth)] and incubated with primary antibodies at 4 °C overnight. For detection, secondary horseradish peroxidase-coupled antibodies were used. For protein detection, Supersignal West Pico Plus chemiluminescent substrate (Thermo Scientific) was used. When necessary, membranes were stripped with stripping buffer containing 0.2 M glycine pH 2.3, 35 mM SDS and 0.1% Tween20 (Roth).

The intensities of the bands were quantified by densitometry using ImageJ. For Fig. EV2C,E,G, the relative pixel densities of the corresponding bands were normalised to the anti- β actin band pixel density. For Fig. EV2G, the relative pixel densities of phosphoproteins between 15 and 250 kDa molecular mass were analysed. For Fig. 2E,G, each dot corresponds to the average of the results of two different technical replicates.

## IL-2 ELISA

In total, 200,000 cells were seeded in each well of an eight-well chambered cover glass (No. 1.5, Cellvis) coated with anti-human CD28 antibody (Biolegend) and Anti-Human CD3 antibody (Invitrogen) and incubated overnight at 37 °C and 5% CO$_2$. Supernatants were harvested, and IL-2 secretion was assessed using the ELISA MAX Deluxe Set Human IL-2 kit (Biolegend) following the manufacturer's instructions. Absorbance of 560 and 405 nm light was measured on the Spark multimode microplate reader (TECAN).

## Labelling for live-cell imaging

For live-cell imaging, cells were labelled with Halo and/or SNAP substrates indicated in the figure legends at a concentration of 1 µM for 1 h at 37 °C in culture media. Cells were washed in growth media at 37 °C for at least 1 h to allow efflux of unbound dye. For imaging cells in resting conditions, 100,000 cells were seeded on 8-well chambered coverglass dishes (No. 1.5, Cellvis).

For imaging of SK-BR-3/Jurkat T-cell interactions, 25,000 SK-BR-3 cells were seeded on 8-well chambered coverglass dishes and cultured overnight. SK-BR-3 cells were stained before imaging with

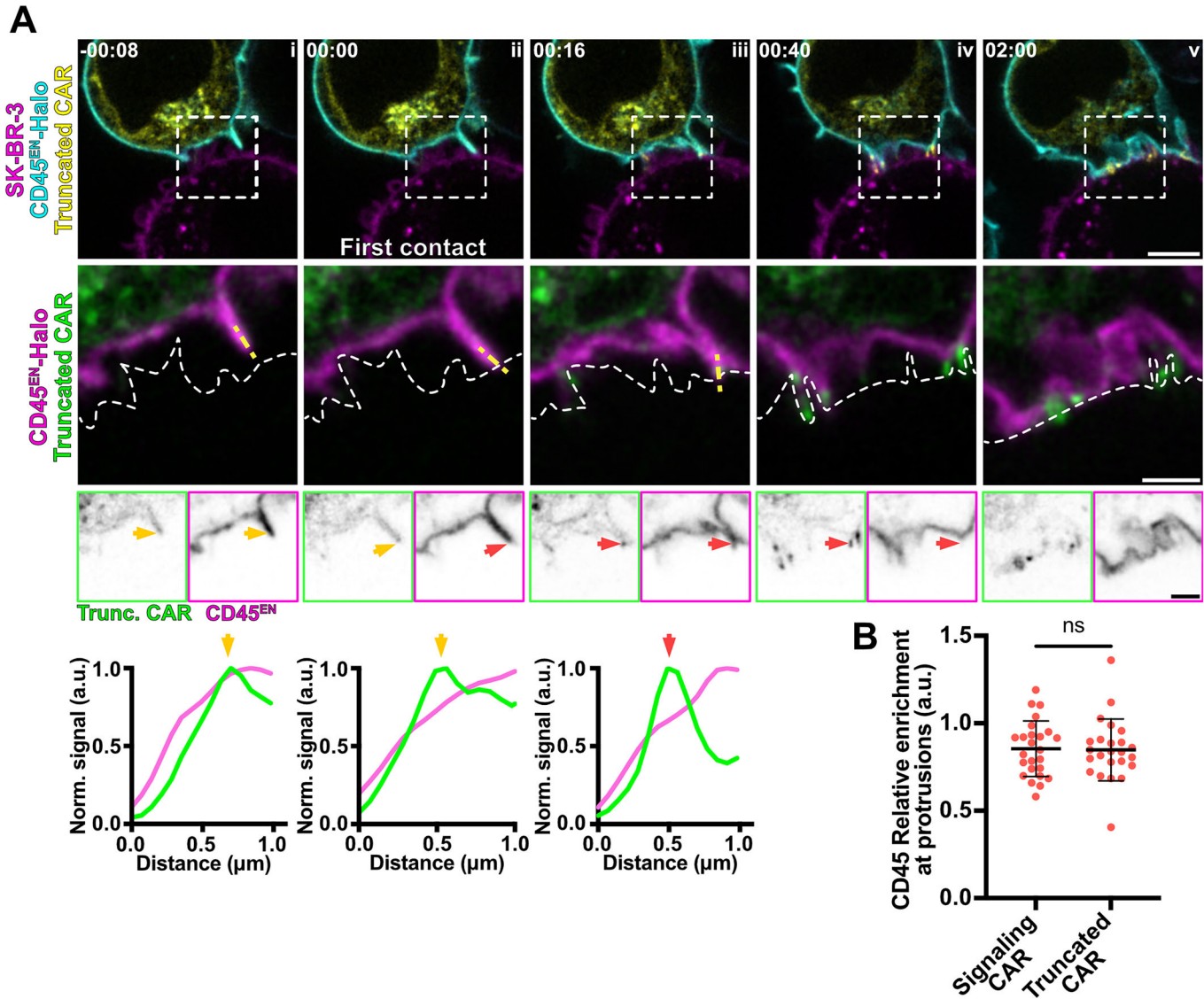

**Figure 7. CD45 is excluded from protrusion contacts with the target cell independently of signalling.**

(A) Time-lapse confocal imaging of Jurkat T cells expressing a truncated HER2-CAR-GFP and CD45EN-Halo labelled with CA-JFX650 interacting with a SK-BR-3 cell stained with CellMaskOrange. The dashed line describes the outline of the SK-BR-3 cell. The truncated HER2-CAR (Trunc. CAR) lacks the co-stimulatory and signalling domains. Yellow arrows highlight a protrusion displaying non-preferential localisation of CD45 or the truncated CAR prior to contact with the target cell, and red arrows highlight a membrane domain with decreased CD45 signal and truncated CAR enrichment. Line profiles show truncated HER2-CAR and CD45 signal along the dashed yellow line. (B) Mean *relative enrichment* of CD45 in protrusion contacts during the first minute of interaction with the target cell mediated by the signalling CAR or by the truncated HER2-CAR. (Signalling HER2-CAR experiment) $n = 18$ independent cell–cell interaction events (truncated HER2-CAR) $n = 24$ independent cell–cell interaction events. P value of unpaired $t$ test is 0.8954. CA chloroalkane (HaloTag substrate. Scale bars, 5 μm (confocal overviews), 2 μm (crops).

the CellMask Orange plasma membrane stain (Invitrogen) for 5 min at 37 °C. In total, 100,000 Jurkat T cells expressing the HER2-CAR-GFP and the protein of interest were added to the wells and imaged immediately after. Live-cell imaging was performed in live-cell imaging solution [FluoBrite DMEM (Gibco) supplemented with 10% FBS, 20 mM HEPES (Gibco) and 1x GlutaMAX (Gibco)].

### Imaging and image processing

Microscopy data were collected on an expert line Abberior STED microscope using the Inspector software from Abberior Instruments (Version 16.3). The microscope is equipped with 485 nm, 561 nm and 640 nm excitation lasers. For two-colour STED experiments, both JF571 (Grimm et al, 2020) and JFX650 (Grimm et al, 2021) dyes were depleted with a 775 nm depletion laser. The detection windows were set to 571–630 nm and 650–756 nm. Multi-colour STED images were recorded line by line sequentially. For three-colour confocal imaging, detection windows were set to 498–540, 590–625 and 660–757 nm. If required for quantitative analysis, laser power was kept constant between images. For live-cell confocal imaging, the pixel size was set to 70 nm and for STED imaging to 30 nm. Live-cell imaging was

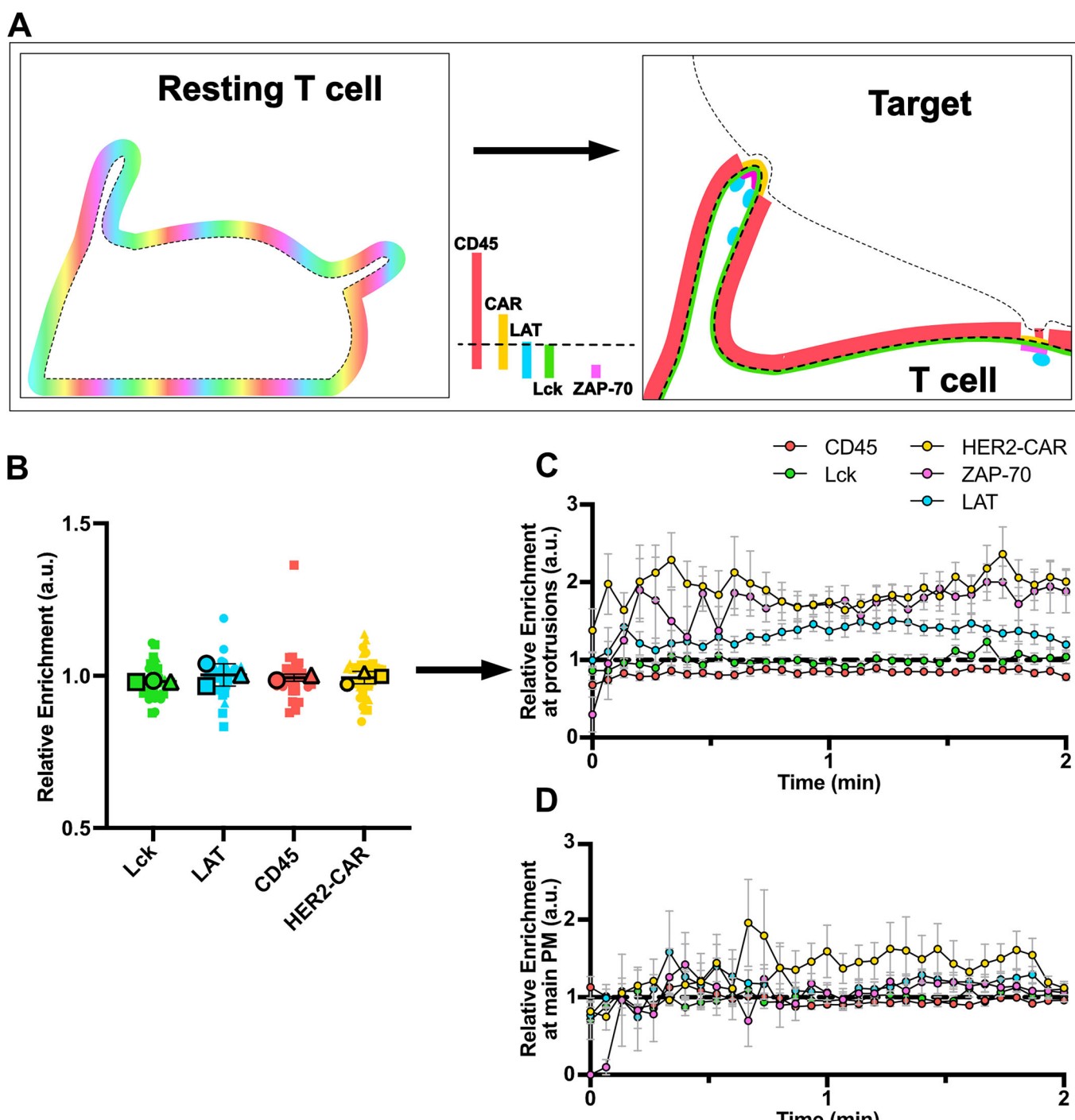

**Figure 8. A quantitative framework of the dynamic rearrangement of signalling proteins on the T cell surface.**

(**A**) Schematic representation of the localisation of the HER2-CAR, Lck, ZAP-70, LAT and CD45 before and at early contact with the target cell. CD45 is excluded from plasma membrane regions at close cell–cell contacts formed both at protrusions and the main body of the T cell, with exclusion being more pronounced at protrusions. This promotes enhanced CAR clustering and activation in protrusions, shown by earlier and greater signal increase of CAR, ZAP-70, and LAT within protrusions at contact sites. (**B**) Quantification of the *relative enrichment* of Lck, LAT, CD45 and the HER2-CAR in resting conditions. Results around 1 indicate no localisation preference to protrusions in comparison to the membrane marker. In total, 36 cells (for HER2-CAR), 32 cells (for Lck), 34 cells (for LAT) and 34 cells (for CD45) from three independent experiments were analysed. (**C, D**) *Relative enrichment* of Lck, LAT, CD45 and the HER2-CAR and *relative membrane-bound ZAP-70* in clusters at (**C**) protrusion membranes and (**D**) main body membrane contacts. $t = 0$ min indicates the first contact detected with the target cell. In total, $n = 49$ cell–cell interactions (for HER2-CAR), $n = 23$ cell–cell interactions (for Lck), $n = 9$ cell–cell interactions (for ZAP-70), $n = 22$ cell–cell interactions (for LAT) and $n = 34$ cell–cell interactions (for CD45) are plotted. Graphs show mean values, s.e.m. error bars.

performed at 37 °C. Both confocal and STED imaging were performed in a single Z-slice.

To reduce noise, confocal images were background subtracted (rolling ball algorithm, radius of 50 pixels) and Gaussian-blurred ($\sigma = 1$ pixel) using Fiji (ImageJ 2.7.0, (Schindelin et al, 2012)). STED images in Figs. 1D, 3F, 4D and 5E were deconvolved using Richardson–Lucy deconvolution implemented in the Python microscopy PYME package (https://python-microscopy.org). Line profiles shown in Figs. 3E, 4D and 5F were obtained using Fiji by drawing a line (with 3-pixel width) alongside the protrusion membrane on Gaussian-blurred ($\sigma = 1$ pixel) STED images. Line profiles on Figs. 5A and 6A were obtained by drawing a line (with 3-pixel width) alongside the protrusion on background-subtracted (rolling ball algorithm, radius of 50 pixels) and Gaussian-blurred ($\sigma = 1$ pixel) confocal images. The line profile data were then normalised and plotted using GraphPad Prism (GraphPad Software, https://www.graphpad.com).

## Image quantification

### Segmentation of protrusions and the main body membrane

Segmentation and quantification of Jurkat T-cell membrane structures were performed using a custom image analysis pipeline (Appendix Fig. S1) implemented in CellProfiler (Stirling et al, 2021). 2-channel pictures of the Jurkat T cell membrane were decomposed into individual .tiff files using Fiji and loaded into CellProfiler. The two channels were then averaged and blurred with a Gaussian filter ($\sigma = 2$ pixels). Segmentation on data from Fig. 3B was performed utilising only the HER2-CAR-GFP channel. Segmentation continued in two parallel streams: identification of the inner body/neighbourhood and identification of membrane structures. For inner body detection, dark holes from the membrane image were enhanced using the rolling ball algorithm in the inverted image. These were thresholded and filtered by area, and internal holes were filled. Frames in which inner body detection was unsuccessful due to high intracellular signal were discarded. The resulting inner body image was dilated by 9 pixels to cover the main body membrane, and by 35 pixels to define the cell neighbourhood. For membrane detection, neurite-like structures were enhanced using the "tubeness" enhancement method and then thresholded. To obtain initial protrusion candidates, the 9-pixel-dilated inner body mask was subtracted from the thresholded neurite structures; negative values were set to 0. The final protrusion mask was obtained by filtering out structures outside of the neighbourhood size, filtering for discarding single-pixel objects. The main body membrane mask was obtained by subtracting the inner body and the protrusion masks from the thresholded neurites. The whole membrane mask was obtained by merging the protrusion and the main body membrane masks.

### Segmentation of membrane contacts

To distinguish between target-contacting and non-contacting regions of the Jurkat T-cell membrane, SK-BR-3 cell masks were first generated with Fiji. The SK-BR-3 channel was background subtracted (rolling ball radius = 50 pixels), and Gaussian-blurred ($\sigma = 2$ pixels). and thresholded with the "Moments" algorithm applied to the last frame of the stack. The SK-BR-3 mask, along with the HER2-CAR and protein of interest channels were loaded

into CellProfiler. Protrusions and main body membranes were segmented as described above.

Contacting regions were determined by applying an "and" operation between SK-BR-3 and the Jurkat T-cell protrusions or main body membrane masks. Non-contacting regions were obtained by subtracting the contacting masks from their respective full masks.

To estimate the contribution of both membranes to the contact with the target in Fig. 2C, Jurkat T cells expressing HER2-CAR-GFP and SNAP-CAAX were imaged interacting with SK-BR-3 cells at a frame rate of 15/s per frame. The areas occupied—in pixels—by the protrusion and main body membrane contacts were quantified and expressed as percentages of total contact area.

### Segmentation of CAR clusters

To segment CAR clusters, the HER2-CAR-GFP channel was normalised with its mean intensity across the whole plasma membrane. The normalised image was then thresholded to obtain all pixels with a value above 2. Size filtering was applied to obtain all structures between the [5–100] area range. Clusters at the main body membrane or protrusion membranes were obtained by applying an "AND" operation between the segmented clusters mask and the protrusion or main body membrane masks.

### Enrichment, relative enrichment and membrane-bound protein

Signal intensity values for each protein were normalised to their maximum values across the whole plasma membrane mask. The parameter "enrichment" (Appendix Fig. S1B) of a protein in a membrane region was defined as

$$\text{Enrichment (protein)} = \frac{\text{Mean Signal (membrane region)}}{\text{Mean signal (whole plasma membrane)}}$$

For Fig. 6D, CD45-to-CAR ratio was calculated as CD45 *enrichment* divided by CAR *enrichment* at CAR clusters; *enrichment* was calculated as described above. For Fig. 1D, dual-colour confocal images of Lck[EN]-Halo/SNAP-CAAX, LAT[EN]-Halo/SNAP-CAAX and CD45[EN]-Halo/SNAP-CAAX were used, and the *relative enrichment* (Appendix Fig. S1B) was computed as

$$\text{Relative Enrichment (protein)} = \frac{\text{Enrichment (protein)}}{\text{Enrichment (SNAP} - \text{CAAX)}}$$

For Fig. 2E, three-colour confocal time lapses of a HER2-CAR-GFP/SNAP-CAAX-expressing Jurkat T cell interacting with a SK-BR-3 cell were obtained with a frame rate of 4 s/frame. Time zero was set as the first frame in which a contact was detected by the pipeline. The CAR *relative enrichment* was computed as above. Figure 2D reports the average of the first 2 min using data collected at both 4 s and 15 s intervals.

For Figs. 4B, 5B, 5C, 6B, 6C and 7B, Jurkat cells expressing HER2-CAR-GFP or the truncated HER2-CAR and the endogenously tagged protein of interest were imaged during interactions with SK-BR-3 cells at 4 s/frame. For Figs. 4B and 6B, the *relative enrichment* of the protein in a membrane contact region (either protrusion or main body membrane) was computed as

$$\begin{aligned} &\text{Relative Enrichment (protein at contact set)} \\ &= \frac{\text{Enrichment (region in contact)}}{\text{Enrichment (region not contacting)}} \end{aligned}$$

For Figs. 4B, 6C and 7B, the mean *relative enrichment* of the first minute of Jurkat T cell/target interaction was calculated. For Fig. 5B, *enrichment* of Lck in the segmented membranes was computed as above. In Fig. 5C, the mean *enrichment* of Lck averaged within the first minute of Jurkat T cell/target interaction was calculated.

For Fig. 3B,C, *membrane-bound* ZAP-70[EN]-Halo was calculated as the difference of ZAP-70 enrichment in contact minus non-contacting regions. This was performed for protrusions and main body membranes separately as their apparent brightness is different when imaged with confocal microscopy. Next, *relative membrane-bound* ZAP-70[EN]-Halo was calculated as shown, employing the mean SNAP-CAAX enrichment obtained in the data set from Fig. 2 as

$$\text{Relative Membrane} - \text{bound} = \frac{\text{Membrane} - \text{bound} (ZAP - 70)}{\text{Enrichment} (SNAP - CAAX)}$$

Uncertainties were propagated accordingly. For Fig. 3C,D, the mean *relative membrane-bound* ZAP-70 for the specified membrane regions during the time interval [0, 60] s was calculated. For Fig. 3E, the ZAP-70-to-CAR ratio at CAR clusters was calculated as the division between membrane-bound ZAP-70 and CAR enrichment at the cluster.

## Statistics and reproducibility

Graphpad Prism 9.3.0 was used to generate all graphs and to perform statistical analysis. Data distribution was assumed to be normal, but not formally tested. Data sets containing continuous data from different biological replicates (Figs. 1C, 1E, EV2A,C,E,F and EV3B) were presented as superplots (Lord et al, 2020). All statistical tests used are indicated in the figure legends. No statistical method was used to predetermine sample size, and sample sizes are indicated in the figure legends. The experiments were not randomised. The Investigators were aware of the group allocation during experiments and outcome assessment. All schematics were generated with Affinity Designer. Microscopy images and Western Blots are shown as representative images.

# Data availability

All data supporting the findings of this study is available at https://www.ebi.ac.uk/biostudies/studies/S-BSST2689.

The source data of this paper are collected in the following database record: S-SCDT-10_1038-S44318-026-00773-5.

# Peer review information

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

## Acknowledgements

This project was supported by the Human Frontier Science Program early-career grant (RGY0088/2021) for XS and FB. Additional funding support for XS includes the American Cancer Society Research (Grant 135926), the NIGMS MIRA program (R35 GM138299), the Gabrielle's Angel Foundation Medical Research Award and the Pershing Square Sohn Prize for Young Investigators in Cancer research. CR was supported by a PhD fellowship from SFB958, Deutsche Forschungsgemeinschaft (DFG). CR and AZ were supported by the DFG – Project Number 278001972 – TRR 186. We thank Steffen Gottshalk for providing the HER2-CAR sequence employed in this study. We thank Luke Lavis (Janelia Research Campus) for providing dyes for live-cell imaging. We thank Yvonne Weber (Freie Universität Berlin) for help with FACS. We thank Petia Adarska for help with imaging and for her valuable comments on the manuscript. We are grateful to Wenting Zhao and Xiangfu Guo (Nanyang Technological University) for the engaging and constructive discussions. We thank Steffen Erdle for cloning the HER2-CAR-Halo plasmid. We thank Lucie Hortmann for valuable support in experiments.

## Author contributions

**Carmen Rodilla-Ramirez**: Conceptualisation; Resources; Data curation; Software; Formal analysis; Supervision; Validation; Investigation; Visualisation; Methodology; Writing—original draft; Writing—review and editing.
**Giorgia Carai**: Data curation; Validation; Investigation; Visualisation; Writing—review and editing. **Eleanor Fox**: Investigation; Methodology; Writing—review and editing. **Amin Zehtabian**: Software; Formal analysis; Writing—review and editing. **Helen Adam**: Data curation; Investigation; Writing—review and editing. **Katja Dallio**: Data curation; Investigation; Writing—review and editing. **Pia Lazki-Hagenbach**: Formal analysis; Writing—review and editing. **Helge Ewers**: Software; Formal analysis; Supervision; Funding acquisition; Writing—review and editing. **Xiaolei Su**: Conceptualisation; Supervision; Funding acquisition; Writing—review and editing. **Francesca Bottanelli**: Conceptualisation; Resources; Data curation; Formal analysis; Supervision; Funding acquisition; Validation; Investigation; Visualisation; Methodology; Writing—original draft; Project administration; Writing—review and editing.

Source data underlying figure panels in this paper may have individual authorship assigned. Where available, figure panel/source data authorship is listed in the following database record: biostudies:S-SCDT-10_1038-S44318-026-00773-5.

## Funding

## Disclosure and competing interests statement

The authors declare no competing interests.

# Expanded View Figures

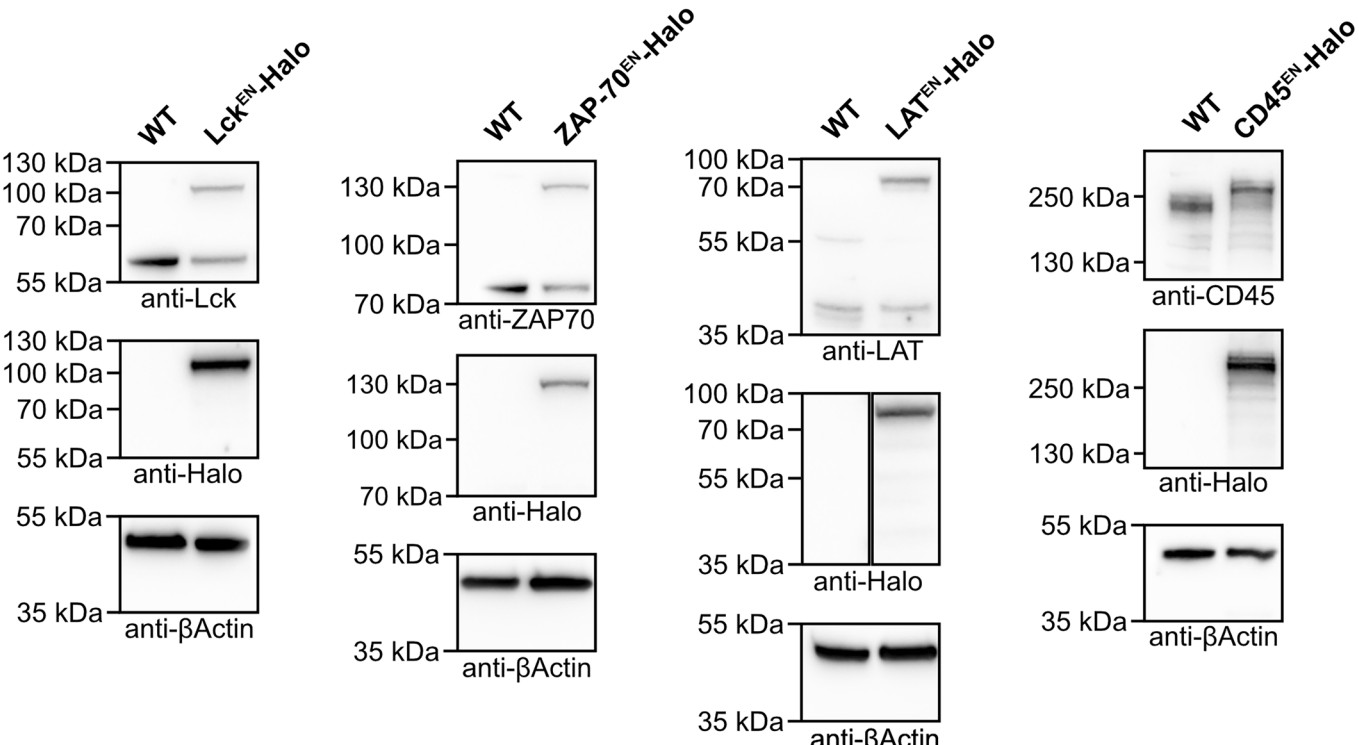

**Figure EV1.  Validation of knock-in (KI) cell lines via western blot.**

Western blots of lysates of Jurkat T cell lines expressing endogenously Halo-tagged Lck, ZAP-70, LAT or CD45. Primary antibodies used for each immunoblot are shown below each crop. Full blots are provided in the source data file. All fusion proteins display the correct shift in molecular weight, corresponding to the molecular weight of the linker (short GS linker for Lck and LAT; long 70aa LAP linker for CD45 and ZAP-70), Halo and an epitope tag (2xalfa or 2xHA). For the CD45 KI, all alleles appear to be edited as shown by a full shift of the band corresponding to WT CD45. For the Lck, ZAP-70 and LAT KI cell lines, only ~50% of alleles are edited.

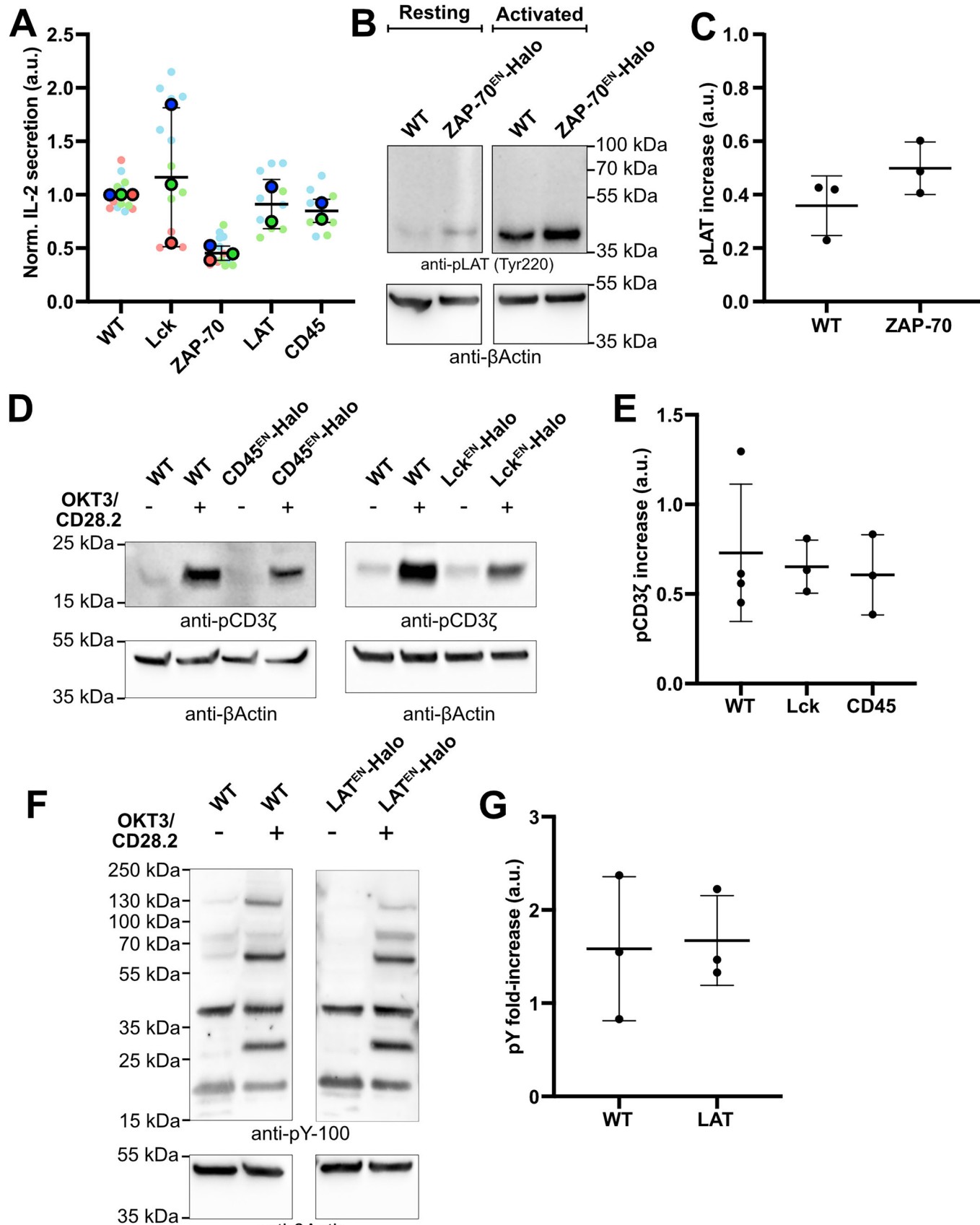

◄

**Figure EV2.   Validation of KI cell lines functionality.**

(**A**) IL-2 secretion was assessed in supernatants of WT and KI Jurkat T cells cultured in dishes coated with OKT3 and CD28.2 antibodies for 24 h. IL-2 secretion is normalised to the mean secretion from unmodified Jurkat T cells from the same day. Replicates are shown in different colours, and each small dot represents the normalised concentration of IL-2 collected from the supernatant of one well. $n = 12$ (WT), $n = 14$ (Lck), $n = 13$ (ZAP-70), $n = 10$ (LAT) and $n = 10$ (CD45) from three (WT, Lck, ZAP-70) or two (LAT, CD45) independent experiments. The graph shows mean values and s.d. error bars. *P* values from paired *t* tests are 0.7050 (WT/Lck), 0.050 (WT/ZAP-70), 0.6840 (WT/LAT) and 0.2970 (WT/CD45). (**B**) Cell lysates derived from the ZAP-70$^{EN}$-Halo cell line or the wildtype Jurkats either resting or activated in OKT3/CD28.2 coated dishes were immunoblotted using anti-pLAT and anti-β-actin antibodies. (**C**) Increase of pLAT signal normalised to β-actin signal upon activation. $n = 3$ experiments are plotted. The graph shows mean values and s.d. error bars. *P* value from paired *t* test is 0.2219. (**D**) Cell lysates derived from the WT, Lck$^{EN}$-Halo and CD45$^{EN}$-Halo, either resting or activated in OKT3/CD28.2 coated dishes, were immunoblotted using anti-pCD3ζ and anti-β-actin antibodies. (**E**) Increase pCD3ζ signal normalised to β-actin signal upon activation. $n = 3$ experiments are plotted. The graph shows mean values and s.d. error bars. *P* values from paired *t* tests are (WT/Lck) 0.7159 and (WT/CD45) 0.5158. (**F**) Cell lysates derived from the WT and LAT$^{EN}$-Halo, either resting or activated in OKT3/CD28.2-coated dishes, were immunoblotted using anti-phosphotyrosine (pY) and anti-β-actin antibodies. (**G**) Increase pY signal normalised to β-actin signal upon activation. $n = 3$ experiments are plotted. The graph shows mean values and s.d. error bars. *P* value from paired *t* test is 0.8747.

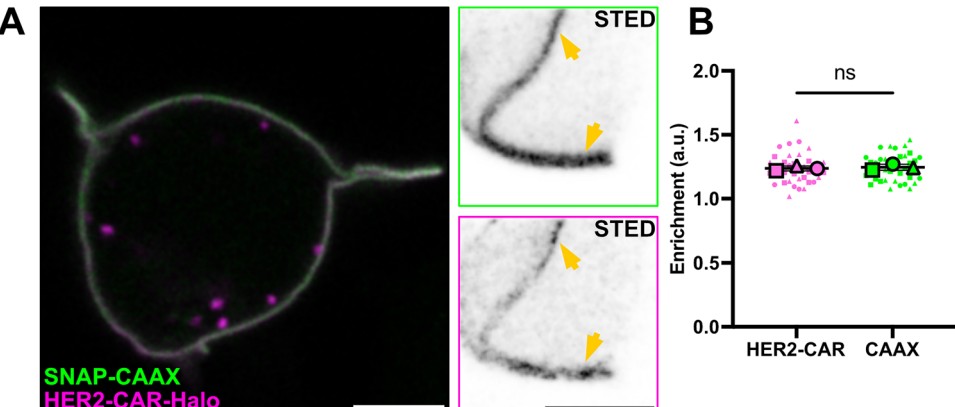

**Figure EV3. HER2-CAR shows no preferential localisation to actin protrusions in resting conditions.**

(A) Live-cell confocal (magenta and green) and STED images (inverted greyscale) of a Jurkat T cell expressing HER2-CAR-Halo labelled with CA-JFX$_{650}$ and SNAP-CAAX labelled with BG-JF$_{571}$. Arrows highlight small HER2-CAR clusters localised either to the main body membrane or to an actin protrusion. (B) *Enrichment* of HER2-CAR tagged with Halo in protrusions. In total, $n = 36$ cells from three independent experiments were analysed. Replicates are shown as different shapes, and each small dot represents a single cell. The graph shows mean values and s.d. error bars. *P* value of paired *t* tests is 0.6116. CA chloroalkane (HaloTag substrate), BG benzylguanine (SNAP-tag substrate). Scale bars, 5 μm (confocal overview), 2 μm (STED images).

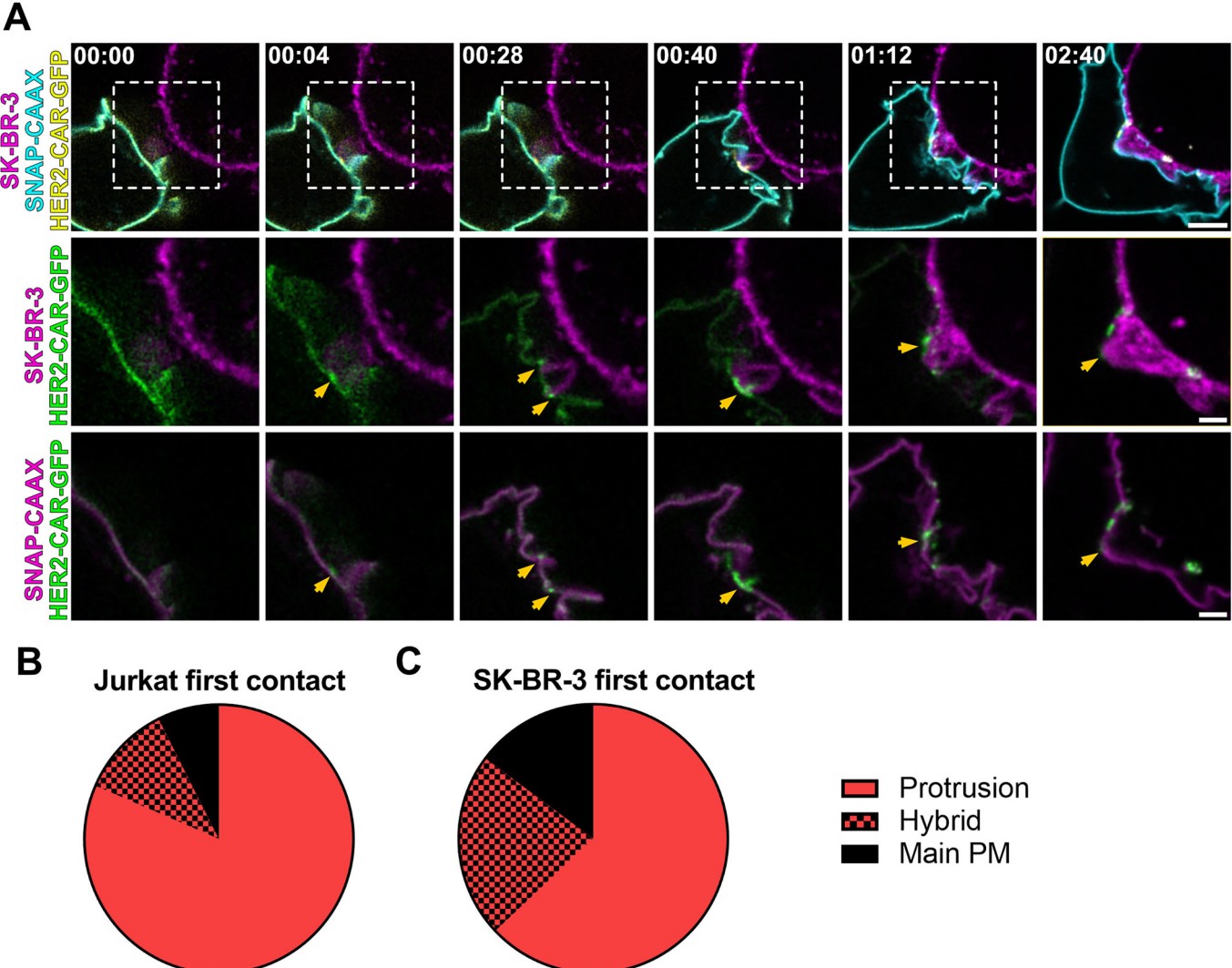

**Figure EV4.  Protrusions from the target cell make contact with Jurkat T cells and lead to successful CAR engagement and clustering.**

(**A**) Time-lapse confocal imaging of a Jurkat T cell expressing HER2-CAR-GFP, SNAP-CAAX (labelled with BG-JF$_{571}$) interacting with a SK-BR-3 cell (labelled with CellMaskOrange). Arrows highlight the cell–cell contact mediated by a protrusion emanating from the SK-BR-3. The contact leads to successful clustering of the HER2-CAR on the Jurkat T-cell membrane. (**B**) Number of Jurkat T cell/SK-BR-3 first detected contacts mediated through Jurkat T-cell protrusions, main body membrane or both at the same time. $n = 29$ from three independent experiments are represented. (**C**) Number of Jurkat T cell/SK-BR-3 first detected contacts mediated through SK-BR-3 protrusions, main body membrane or both at the same time. $n = 29$ independent cell–cell interaction events were analysed. BG benzylguanine (SNAP-tag substrate). Scale bars, 5 µm (confocal overviews), 2 µm (crops).

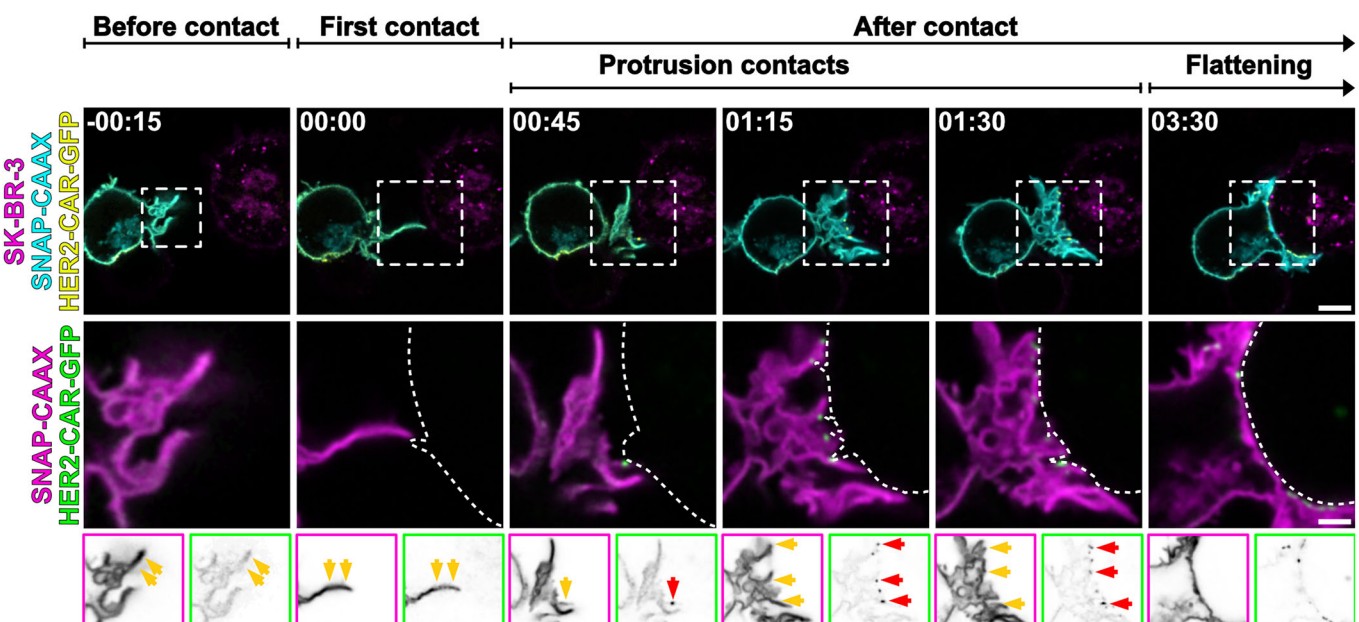

**Figure EV5.   Early contact between a CD4+ primary T cell and the target cell is mediated by protrusions that trigger CAR clustering and activation.**

Time-lapse confocal imaging of a CD4+ human T cell expressing HER2-CAR-GFP, SNAP-CAAX (labelled with BG-JF$_{571}$) interacting with a SK-BR-3 cell (labelled with CellMaskOrange). The dashed line describes the outline of the SK-BR-3 cell. HER2-CAR shows no preferential localisation to protrusions prior to first detected contact with the target at $t = 0$ s, but is enriched in protrusion contacts. BG benzylguanine (SNAP-tag substrate). Scale bars, 5 μm (confocal overviews), 2 μm (crops).

