## [Peer Review File · The EMBO Journal]

T-cell protrusions enable fast, localised initiation of chimeric antigen receptor signalling

Carmen Rodilla-Ramirez, Giorgia Carai, Eleanor Fox, Amin Zehtabian, Helen Adam, Katja Dallio, Pia Lazki-Hagenbach, Helge Ewers, Xiaolei Su, and Francesca Bottanelli

Corresponding authors: Francesca Bottanelli (francesca.bottanelli@fu-berlin.de) , Carmen Rodilla-Ramirez (carmenair97@zedat.fu-berlin.de)

Review Timeline:

Submission Date:	22nd Jul 25
Editorial Decision:	10th Sep 25
Revision Received:	4th Feb 26
Editorial Decision:	27th Feb 26
Revision Received:	11th Mar 26
Accepted:	26th Mar 26

Editor: Ieva Gailite

Transaction Report:

Dear Dr. Bottanelli,

Thank you for submitting your manuscript for consideration by the EMBO Journal. We have now received comments from two reviewers, which are included below for your information. Since the third reviewer was not able to submit their report in time, I am taking the decision based on the evaluations at hand.

As you will see, both reviewers are generally positive in their evaluation and appreciate the contribution of the study to the research field. However, they also indicate a number of generally reasonable and overlapping concerns that would need to be addressed before they can support publication here. Based on these positive assessments, I invite you to submit a revised manuscript in response to the comments by all reviewers. I would also be happy to discuss the revision in more detail via email or phone/videoconferencing.

We generally allow three months as standard revision time, which can be extended to six months in the case of major revisions. Should you foresee a problem in meeting this deadline, please let us know in advance to discuss an extension.

As a matter of policy, competing manuscripts published during this period will not negatively impact on our assessment of the conceptual advance presented by your study. However, please contact me as soon as possible upon publication of any related work to discuss the appropriate course of action.

When preparing your letter of response to the referees' comments, please bear in mind that this will form part of the Review Process File and will therefore be available online to the community. For more details on our Transparent Editorial Process, please visit our website: <https://www.embopress.org/page/journal/14602075/authorguide#transparentprocess>. Please also see the attached instructions for further guidelines on preparation of the revised manuscript.

Please feel free to contact me if you have any further questions regarding the revision. Thank you for the opportunity to consider your work for publication. I look forward to your revision.

With best wishes,

Ieva

- a point-by-point response to the referees' comments, with a detailed description of the changes made (as a word file).
- a word file of the manuscript text.
- individual production quality figure files (one file per figure)
- a complete author checklist, which you can download from our author guidelines (<https://www.embopress.org/page/journal/14602075/authorguide>).

- Expanded View files (replacing Supplementary Information)

- a Reagents and Tools Table as part of the Methods section, which can be downloaded from our author guidelines

(<https://www.embopress.org/page/journal/14602075/authorguide#structuredmethods>)

We realize that it is difficult to revise to a specific deadline. In the interest of protecting the conceptual advance provided by the work, we recommend a revision within 3 months (9th Dec 2025). Please discuss the revision progress ahead of this time with the editor if you require more time to complete the revisions.

Referee #1:

General Summary and Opinion

The initiation of contact between T cells and antigen-presenting cells (including tumor cells) is a critical step in T cell-mediated immune responses. Therefore, gaining greater insight into how T cells organize their membrane topology and immune signaling apparatus during this process is important for our understanding of adaptive immune function as a whole. Importantly, these interactions are thought to initiate most frequently via actin-rich protrusions of the T cell membrane. Rodilla-Ramirez et al note that previous studies investigating the organization of T cell signaling components within actin-rich protrusions have produced inconsistent results and rely on methods, such as tagged protein overexpression and antibody staining of fixed cells, that may disrupt or mis-represent endogenous localization. The authors sought to enhance our understanding of the spatial relationship between the T cell receptor and proximal signaling components by combining live, STED-based super-resolution imaging with endogenous tagging of signaling proteins using CRISPR. Using Jurkat T cells expressing a chimeric antigen receptor (CAR), the authors conclude that, in contrast to previous studies, none of the signaling components studied (CAR, Lck, LAT, CD45, and Zap-70) are enriched in membrane protrusions prior to contact. Upon contact initiation, the CAR, LAT, and Zap-70 all rapidly accumulated at the site of interaction. Conversely, CD45 was quickly excluded from the interaction site and Lck distribution showed no discernible change in distribution during contact and synapse formation.

This study addresses an important controversy in the field which has relevance for the development of future CAR T cell therapies. The authors' conclusions challenge accepted models of T cell signaling component organization and present a novel platform to study this organization with high spatial and temporal resolution. However, the study would be strengthened by a more in-depth confirmation of the functionality of the tagged proteins and the manuscript would benefit from a discussion more focused on the relevance of the observations for the function of CAR-expressing cells in particular.

Specific Major Concerns

The authors conclusions rely heavily on the assumption that the tagged proteins localize and function normally despite the fact that the Halo tag nearly doubles the size of the protein in some instances. In my opinion, the IL-2 secretion data included is not sufficient to confirm the proper functioning and localization of these constructs. Indeed, the Zap-70 knock-in cells appear to produce roughly half the IL-2 of WT counterparts (a very consistent result, though statistical significance was not reached). We suggest two relatively straightforward ways to make clear that the functionality and localization of the tagged constructs is not perturbed by the tagging. First, a western blot can confirm that signals directly downstream of the tagged effectors are fully intact. For example, assessing CAR ITAM and/or Zap-70 phosphorylation downstream of the tagged Lck, or assessing LAT phosphorylation downstream of tagged Zap-70 would hopefully show that the tagged proteins are able to successfully transduce signaling. Similarly, measuring global phosphotyrosine levels upon activation would be a simple and useful readout. Second, dynamic imaging of knock-in cells on a stimulatory substrate should help determine whether the tagged proteins can interact with antigen receptor clusters as previously described by other groups. In particular, measurements of the kinetics of the recruitment of Halo-tagged LAT, Zap-70, and Lck to antigen receptor clusters in comparison to previous studies would hopefully remove any doubts about whether the tagged constructs interact with native signaling components normally.

It is not clear from the text whether imaging data were collected as a single Z-slice, or if 3D imaging data were obtained. However, all images are presented as a single Z-slice. From this, it is difficult to confirm that the region labeled the "tip" of a protrusion is indeed the "tip" and does not, in fact, extend further in Z. This also makes it difficult to confidently identify the initial contact, as contact may have already occurred in another imaging plane. In either case, it would be beneficial for the authors to

describe the imaging methodology in greater detail to clarify this, and possibly to include a description of how initial contacts were identified.

Minor Concerns

The text of the manuscript indicates that the sequences associated with the Cas 9 guide RNA are included in "Supplementary Table 3." However, no such table was included in the supplemental material. Please ensure that this information is included. It may be beneficial to further describe the design of the tagged constructs, especially if steps were taken to ensure that the Halo tag would not affect function or localization. For example, if previous studies have utilized a similar tagging strategy and confirmed its effectiveness, please consider adding citations to this effect.

The study centers around the use of the HER2-targeting CAR construct as the primary antigen receptor. This is a very relevant system, in particular given the importance of understanding T cell signaling in the context of CAR T cell therapy. However, the authors could make this distinction more clearly in both the introduction and discussion. Many studies have considered the differences in signaling quality and synapse organization between conventional TCRs and CARs. Elaborating on this context would strengthen the central argument of the report.

The lack of measurable Lck enrichment at contact sites is somewhat surprising. Is it possible that, since the CAR does not rely on CD4 co-receptor binding, Lck is a more minor component of a CAR signaling apparatus than with a conventional TCR? Functional studies of the tagged Lck and/or a discussion of the possible differences between CARs and conventional TCRs in their utilization of Lck may provide clarity.

Additional Suggestions

As a control, it may be beneficial to perform experiments with tagged constructs intentionally mutated to abolish functionality or localization. This would confirm that canonically important motifs are still driving function and localization in the tagged constructs.

As stated previously, the manuscript would be enhanced by a greater discussion of the known and possible differences between conventional TCRs and CARs in the processes explored here. In addition, expressing a conventional TCR in the knock-in Jurkat cells would control for these differences and may reveal interesting specificities in the signaling component organization and dynamics.

The authors perform one experiment with primary CD4⁺ T cells. However, given the focus on the HER2-CAR, it might be more relevant to examine the response of primary CD8⁺ T cells, since actual CAR therapies are more likely to utilize CD8⁺ T cells.

The formatting of Supplementary Table 1 appears to be distorted. Please consider altering the size or orientation to ensure that the full table appears together and intact on the page.

In paragraph 4 of the Discussion, the authors state that the "primary mechanism by which actin-rich protrusion (sic) promote increased CAR accumulation..." is through the exclusion of CD45. However, the authors do not address this dynamic directly and cannot definitively draw this conclusion. I recommend re-framing this statement for greater accuracy.

Referee #2:

This paper develops a quantitative super-resolution imaging-based framework to explain how actin-rich protrusions on T cells initiate CAR-mediated immune signalling. Using CRISPR knock-in lines with endogenously tagged proteins and live-cell super-resolution imaging, the authors show that key signalling components such as Lck, LAT, and CD45 are not pre-enriched in protrusions at rest. Instead, upon contact with HER2-positive target cells, CARs rapidly cluster within protrusions, accompanied by fast recruitment of ZAP-70 and LAT and efficient exclusion of CD45, while Lck remains evenly distributed. These findings establish protrusions as privileged sites for the earliest steps of CAR activation, where their geometry and capacity for CD45 exclusion create a nanoscale environment that accelerates signalling and synapse formation.

Within the CAR and TCR signalling fields, the importance of this study is high. The core novel finding is that signalling proteins are not pre-positioned in protrusions but are dynamically recruited there upon contact, with CD45 exclusion preceding CAR clustering. This departs from some earlier claims of pre-enrichment and reframes protrusions as sites of enhanced activation by exclusion and clustering dynamics rather than by pre-loaded signalling machinery. The novel imaging approach taken by the authors to characterising the role of microvilli contacts helps clarify and reconciles previous contradictory reports. Perhaps the only downside is that preclustering of the TCR on the tips of microvilli was not tested, which would have helped generalised comparisons to other studies that used the TCR. I appreciate that this is outside the focus of study however.

In general the conclusions are well supported by very high-quality data. The weakest point is perhaps generalisation from CAR T cell-APC contacts to TCR activation in general, since CAR-mediated contacts are less stable and may exaggerate protrusion contributions.

Some specific comments relating to measuring the signalling protein recruitment:

- The use of the CAAX probe to normalise membrane-bound protein intensity between protrusions and the cell body is a strong control. How was an equivalent normalisation handled for cytoplasmic proteins such as ZAP-70, where membrane area does not provide an appropriate reference?
- An additional and potentially informative metric for assessing signalling efficiency at microvilli would be the ratio of ZAP-70-HALO intensity to CAR-GFP intensity. If microvilli are indeed privileged sites for initiating signalling, one would expect this ratio to be higher at microvilli compared to the cell body.
- A potentially important caveat in the quantification of LAT clusters at microvilli versus the cell body (Fig. 3E) is the effect of optical sectioning. For microvilli, the imaging plane is likely to capture the entire membrane-associated signal, whereas a CAR cluster on the flatter cell body may be associated with a LAT cluster that lies outside the z-plane. This could contribute to the relatively modest, though statistically significant, difference reported.

Minor comment:

- I couldn't find the supplementary movies in the files for reviewers so I could not evaluate these.

Non-essential suggestion:

- There are freely available machine-learning approaches to segmenting cells, such as Ilastik (<https://www.ilastik.org/>), that may allow more nuanced segmentation of different types of protrusions (eg microvilli, filopodia and pseudopodia). It might be informative to make a distinction between the distribution of CAR, Lck and CD45 in the three types.

Answers to referees comments (in red)

Referee #1:

General Summary and Opinion.

The initiation of contact between T cells and antigen-presenting cells (including tumor cells) is a critical step in T cell-mediated immune responses. Therefore, gaining greater insight into how T cells organize their membrane topology and immune signaling apparatus during this process is important for our understanding of adaptive immune function as a whole. Importantly, these interactions are thought to initiate most frequently via actin-rich protrusions of the T cell membrane. Rodilla-Ramirez et al note that previous studies investigating the organization of T cell signaling components within actin-rich protrusions have produced inconsistent results and rely on methods, such as tagged protein overexpression and antibody staining of fixed cells, that may disrupt or mis-represent endogenous localization. The authors sought to enhance our understanding of the spatial relationship between the T cell receptor and proximal signaling components by combining live, STED-based super-resolution imaging with endogenous tagging of signaling proteins using CRISPR. Using Jurkat T cells expressing a chimeric antigen receptor (CAR), the authors conclude that, in contrast to previous studies, none of the signaling components studied (CAR, Lck, LAT, CD45, and Zap-70) are enriched in membrane protrusions prior to contact. Upon contact initiation, the CAR, LAT, and Zap-70 all rapidly accumulated at the site of interaction. Conversely, CD45 was quickly excluded from the interaction site and Lck distribution showed no discernible change in distribution during contact and synapse formation. This study addresses an important controversy in the field which has relevance for the development of future CAR T cell therapies. The authors' conclusions challenge accepted models of T cell signaling component organization and present a novel platform to study this organization with high spatial and temporal resolution.

However, the study would be strengthened by a more in-depth confirmation of the functionality of the tagged proteins and the manuscript would benefit from a discussion more focused on the relevance of the observations for the function of CAR-expressing cells in particular.

Specific Major Concerns.

The authors conclusions rely heavily on the assumption that the tagged proteins localize and function normally despite the fact that the Halo tag nearly doubles the size of the protein in some instances. In my opinion, the IL-2 secretion data included is not sufficient to confirm the proper functioning and localization of these constructs. Indeed, the Zap-70 knock-in cells appear to produce roughly half the IL-2 of WT counterparts (a very consistent result, though statistical significance was not reached). We suggest two relatively straightforward ways to make clear that the functionality and localization of the tagged constructs is not perturbed by the tagging. First, a western blot can confirm that signals directly downstream of the tagged effectors are fully intact. For example, assessing CAR ITAM and/or Zap-70 phosphorylation downstream of the tagged Lck, or assessing LAT phosphorylation downstream of tagged Zap-70 would hopefully show that the tagged proteins are able to successfully transduce signaling. Similarly, measuring global phosphotyrosine levels upon activation would be a simple and useful readout. Second, dynamic imaging of knock-in cells on a stimulatory substrate should help determine whether the tagged proteins can interact with antigen receptor clusters as previously described by other groups. In particular, measurements of the kinetics of the recruitment of Halo-tagged LAT, Zap-70, and Lck to antigen receptor clusters in

comparison to previous studies would hopefully remove any doubts about whether the tagged constructs interact with native signaling components normally.

We agree with the referee, and we thank them for suggesting simple and straight-forward experiments that can easily answer these questions. We have addressed this comment in several ways:

1. By assessing LAT phosphorylation for the ZAP-70^{EN}-Halo cell line with a Western Blot. Immunoblotting (New Figure EV 2B) and quantification (Figure EV 2C) show that the increase of phosphorylated LAT upon activation for the ZAP-70^{EN}-Halo cell line does not differ from the unmodified Jurkat WT cell line.
2. By measuring CD3 ζ phosphorylation for the Lck^{EN}-Halo and the CD45^{EN}-Halo cell lines upon activation. Immunoblotting (Figure EV 2D) and quantification (Figure EV 2E) show no difference in the level of pCD3 ζ upon activation in comparison to the unmodified Jurkat WT.
3. By measuring general phosphotyrosine levels for LAT^{EN}-Halo cell line. Immunoblotting (Figure EV 2F) and quantification (Figure EV 2G) show no significant difference in the increase of general phosphotyrosine levels upon activation in the cell line in comparison to the unmodified Jurkat WT.

In summary, we have included a new figure (Figure EV 2) that shows all the controls performed to test the functionality of the cell lines and the tagged constructs (including the IL-2 assays which were previously included in Supplementary Figure 1). We have re-written the corresponding results section (lines 18-22, page 4). Details about the blot preparation and quantification are now added in the Methods section. We believe that these additional controls further strengthen the idea that the tagged proteins are functional. We reasoned that the western blot experiments were a more robust control in comparison to the imaging experiments suggested by the reviewer as imaging experiments are influenced by many parameters (cell-cell variability, cell type used, activation substrate, etc) and only indirectly measure the phosphorylation (or de-phosphorylation activity) of the tagged proteins.

It is not clear from the text whether imaging data were collected as a single Z-slice, or if 3D imaging data were obtained. However, all images are presented as a single Z-slice. From this, it is difficult to confirm that the region labeled the "tip" of a protrusion is indeed the "tip" and does not, in fact, extend further in Z. This also makes it difficult to confidently identify the initial contact, as contact may have already occurred in another imaging plane. In either case, it would be beneficial for the authors to describe the imaging methodology in greater detail to clarify this, and possibly to include a description of how initial contacts were identified.

The imaging data was collected in a single z-slice. We have now clarified this by explicitly stating it in "Imaging and image processing", the Methods section: "*Both confocal and STED imaging was performed in a single Z-slice.*"

We agree with the reviewer statement that is not possible to distinguish the tip from other parts of the protrusion, and the protrusion could extend further in Z. For this reason, we did not explicitly write protrusion "tips", but only distinguished between protrusive membranes and non-protrusive (main PM) membranes. Our statement on increased clustering and exclusion dynamics in protrusions is not limited to the tips of the protrusions but rather occurs at the level of the whole structure.

In general, contacts between Jurkat and target were identified by applying a logical "AND" operation between the mask of the SK-BR-3 cell and the Jurkat cell membrane. A more detailed description can be found in the "Image Quantification" section, specifically under "Segmentation of membrane contacts".

We agree with the referee that, because of the single-plane imaging, our timelapses cannot fully exclude the possibility that the first contact might have occurred outside of the imaged plane before it was detected in the selected Z-slice. We have assumed that the plane we observe is representative of other planes and that the large number of imaging repetitions strengthens our conclusions that first contacts are typically mediated by protrusions. Additionally, we ensured that the imaged Jurkat cell was sufficiently distant from the target cell at the start of acquisition, allowing us to reasonably assume that no contact had yet occurred in any plane before starting the acquisition.

For clarity and accuracy, we have change “first contact” to “early contact”, “before contact”, or to “first contact detected”.

E.g.

“Live-cell confocal microscopy allowed us to follow cell-cell interactions from the first contact until immunological synapse formation” was changed to “Live-cell confocal microscopy allowed us to follow cell-cell interactions **from before contact** until immunological synapse formation”
“The first contact of the T cell with the target cell usually occurred on actin-rich protrusions” was changed to “The first contact **detected between** the T cell **and** the target cell usually occurred on actin-rich protrusions”

“While CAR clustering (Fig. 5Aiii) is prominent after first contact (Fig. 5Aii), LCK remains evenly distributed on the PM of T cells.” Was changed to “While CAR clustering (Fig. 5Aiii) is prominent **in early contacts** (Fig. 5Aii), LCK remains evenly distributed on the PM of T cells.”

Minor Concerns.

The text of the manuscript indicates that the sequences associated with the Cas 9 guide RNA are included in "Supplementary Table 3." However, no such table was included in the supplemental material. Please ensure that this information is included. It may be beneficial to further describe the design of the tagged constructs, especially if steps were taken to ensure that the Halo tag would not affect function or localization. For example, if previous studies have utilized a similar tagging strategy and confirmed its effectiveness, please consider adding citations to this effect.

We apologise, information on guides sequences is now available in extended view table 3.

We agree with the referee and we have cited work confirming that similarly designed tagged constructs retain functionality.

Lines 22-25, page 4: “This is consistent with previous reports showing that Lck (Ehrlich et al., 2002), ZAP-70 (Sloan-Lancaster et al., 1998) and LAT (Saez et al., 2025) C-terminal tagging with tags of similar size retain protein functionality”

The study centers around the use of the HER2-targeting CAR construct as the primary antigen receptor. This is a very relevant system, in particular given the importance of understanding T cell signaling in the context of CAR T cell therapy. However, the authors could make this distinction more clearly in both the introduction and discussion. Many studies have considered the differences in signaling quality and synapse organization between conventional TCRs and CARs. Elaborating on this context would strengthen the central argument of the report.

We thank the referee for the suggestion. We agree that additional information about CAR T signalling and differences with TCR-dependent signalling would strengthen the paper. We have now introduced CAR T cells in the introduction and mentioned that while signalling machinery between CAR and TCRs are mostly shared, there are qualitative and quantitative differences that differ based on the choice of CD3 or coreceptor domains. Specifically, we have added the following paragraph at the beginning of the introduction (lines 10-17, page 2).

“Researchers have exploited the power of T cells to target tumours that evade the immune system by harming T cells with Chimeric Antigen Receptors (CARs). CARs are synthetic receptors that typically combine antibody-derived ligand-binding motifs with intracellular domains of the TCR and coreceptors (Xiong et al., 2024). CAR signalling harnesses proximal effector molecules similar to those of the TCR, including Src-family kinases, ZAP-70 or LAT, but differs in magnitude and kinetics (Davenport et al., 2018; Gudipati et al., 2020; Karlsson et al., 2015; Salter et al., 2021)”

Moreover, we have commented on a key study investigating the differential role of actin-rich protrusions during CAR vs TCR mediated activation. This study was already mentioned in the discussion (lines 25-28, page 2, changes are shown in bold):

*“Actin-rich protrusions, commonly referred to as T cell microvilli, have been shown to rapidly scan the APC surfaces and stabilise upon TCR-pMHC binding (Cai et al., 2017) **or hyper-stabilise upon CAR-ligand interaction (Beppler et al, 2023)”***

We have added a reference to a publication showing that CD45 exclusion affects CAR signaling (lines 36-40, page 2, changes are shown in bold):

*“Size-dependent exclusion of CD45 from TCR-pMHC **or CAR-ligand** interaction sites, shown at different timepoints of T cell activation (Chang et al., 2016a; Razvag et al., 2018; Varma et al., 2006, **Xiao et al., 2022**), has been proposed to be the driving element for TCR **and CAR** triggering, according to the kinetic segregation model (Davis and van der Merwe, 2006).”*

We have re-written the discussion to include a paragraph discussing our data in the context of TCR and CAR signalling research, including aspects on the differential organization of the IS (lines 18-32, page 9).

“Previous work shows that protrusions are hyper-stabilised during CAR-mediated activation (Beppler et al., 2023) and that this stabilisation depends on ligand density and receptor affinity. Hyper-stabilised signalling foci (as also observed here) could cause multi-focal synapse characteristic of some CARs (Beppler et al., 2023; Davenport et al., 2018; Gudipati et al., 2020; Xiong et al., 2018). In a physiological T cell-APC contact, where pMHC is less dense, and the affinity of the interaction is several orders of magnitude lower, protrusions are likely shorter-lived. Under these conditions, TCR accumulation at protrusions may be considerably less pronounced than what we observe here for the HER2-CAR. As CD45 exclusion is signalling-independent (Figure 7), we speculate that there may be no differences in exclusion dynamics between CAR- and TCR-dependent activation. In addition, the lack of measurable Lck rearrangement in the CAR system (Figure 5) is in agreement with reports of reduced Lck recruitment in mature CAR synapses in comparison to the TCR (Davenport et al., 2018). This could possibly be explained by the absence of CD4-bound Lck brought in proximity of the CAR by MHC-CD4 interactions (Artyomov et al., 2010) or a lower reliance of CD28-based CARs on Lck phosphorylation (Wu et al., 2023).”

The lack of measurable Lck enrichment at contact sites is somewhat surprising. Is it possible that, since the CAR does not rely on CD4 co-receptor binding, Lck is a more minor component of a CAR signaling apparatus than with a conventional TCR?

Functional studies of the tagged Lck and/or a discussion of the possible differences between CARs and conventional TCRs in their utilization of Lck may provide clarity.

We agree with a referee that a discussion on differences on Lck recruitment between TCR- and CAR-dependent signalling may provide clarity on our results. We have changed the discussion accordingly (lines 27-32, page 9, changes in bold).

“In addition, the lack of measurable Lck rearrangement in the CAR system (Figure 5) is in agreements with reports of reduced Lck recruitment in mature CAR synapses in comparison to the TCR (Davenport et al., 2018). This could possibly be explained by the absence of CD4-bound Lck brought in proximity of the CAR by MHC-CD4 interactions (Artyomov et al., 2010) or a lower reliance of CD28-based CARs on Lck phosphorylation (Wu et al., 2023).”

Additional Suggestions.

As a control, it may be beneficial to perform experiments with tagged constructs intentionally mutated to abolish functionality or localization. This would confirm that canonically important motifs are still driving function and localization in the tagged constructs.

We appreciate this comment and will use this feedback for controls in future research projects.

As stated previously, the manuscript would be enhanced by a greater discussion of the known and possible differences between conventional TCRs and CARs in the processes explored here. In addition, expressing a conventional TCR in the knock-in Jurkat cells would control for these differences and may reveal interesting specificities in the signaling component organization and dynamics.

We agree that exploring the role of actin-rich protrusions in TCR-dependent activation is an exciting research avenue that we are currently pursuing. We have expanded both the introduction and discussion sections and elaborated more on the CAR vs TCR systems (see minor point 2 and 3).

The authors perform one experiment with primary CD4⁺ T cells. However, given the focus on the HER2-CAR, it might be more relevant to examine the response of primary CD8⁺ T cells, since actual CAR therapies are more likely to utilize CD8⁺ T cells.

We agree exploring the role of actin protrusions and CAR-dependent signaling in CD8⁺ primary T cells would be interesting. However, we used the model Jurkat T cell, that is a CD4⁺ T cell. Therefore, we carried out experiments with primary CD4⁺ T cells, confirming that results are not restricted to the model cell line.

The formatting of Supplementary Table 1 appears to be distorted. Please consider altering the size or orientation to ensure that the full table appears together and intact on the page.

Thank you for noticing, we have corrected the formatting.

In paragraph 4 of the Discussion, the authors state that the "primary mechanism by which actin-rich protrusion (sic) promote increased CAR accumulation..." is through the exclusion of CD45. However, the authors do not address this dynamic directly and cannot definitively draw this conclusion. I recommend re-framing this statement for greater accuracy.

We have re-written the statements: *“Our data suggests that actin-rich protrusions may enhance CAR accumulation (Figure 2C-D) and activation (Figure 3D) by excluding CD45 from the interaction site more rapidly (Figure 6B) and to a greater extent (Figure 6C-D) than the main body membrane”* (lines 1-4, page 9)

Referee #2:

This paper develops a quantitative super-resolution imaging-based framework to explain how actin-rich protrusions on T cells initiate CAR-mediated immune signalling. Using CRISPR knock-in lines with endogenously tagged proteins and live-cell super-resolution imaging, the authors show that key signalling components such as Lck, LAT, and CD45 are not pre-enriched in protrusions at rest. Instead, upon contact with HER2-positive target cells, CARs rapidly cluster within protrusions, accompanied by fast recruitment of ZAP-70 and LAT and efficient exclusion of CD45, while Lck remains evenly distributed. These findings establish protrusions as privileged sites for the earliest steps of CAR activation, where their geometry and capacity for CD45 exclusion create a nanoscale environment that accelerates signalling and synapse formation.

Within the CAR and TCR signalling fields, the importance of this study is high. The core novel finding is that signalling proteins are not pre-positioned in protrusions but are dynamically recruited there upon contact, with CD45 exclusion preceding CAR clustering. This departs from some earlier claims of pre-enrichment and reframes protrusions as sites of enhanced activation by exclusion and clustering dynamics rather than by pre-loaded signalling machinery. The novel imaging approach taken by the authors to characterising the role of microvilli contacts helps clarify and reconciles previous contradictory reports. Perhaps the only downside is that preclustering of the TCR on the tips of microvilli was not tested, which would have helped generalised comparisons to other studies that used the TCR. I appreciate that this is outside the focus of study however.

In general the conclusions are well supported by very high-quality data. The weakest point is perhaps generalisation from CAR T cell-APC contacts to TCR activation in general, since CAR-mediated contacts are less stable and may exaggerate protrusion contributions.

We thank referee 2 for the encouraging words. We find the statement that our research *“reframes protrusions as sites of enhanced activation by exclusion and clustering dynamics rather than by pre-loaded signalling machinery”* is a nice and concise way of summarising the relevance of our findings. We have therefore rephrased the end of the abstract to include a similar statement (lines 23-25, page 1).

Perhaps the only downside is that preclustering of the TCR on the tips of microvilli was not tested, which would have helped generalised comparisons to other studies that used the TCR. I appreciate that this is outside the focus of study however.

While we agree that this is outside the scope of the current manuscript, we do have exciting preliminary data on the localisation of the TCR within actin-rich protrusions in resting conditions. Interestingly, CD3 ζ localisation is similar to the CAR localisation: strongly pre-clustered on the plasma membrane but not specifically enriched in actin-rich protrusions.

Some specific comments relating to measuring the signalling protein recruitment:
- The use of the CAAX probe to normalise membrane-bound protein intensity between protrusions and the cell body is a strong control. How was an equivalent normalisation handled for cytoplasmic proteins such as ZAP-70, where membrane area does not provide an appropriate reference?

The analysis we applied to ZAP-70 was done with the same parameters as the analysis for the other membrane-bound proteins. This comment motivated us to improve the analysis and to develop a ZAP-70-specific quantification metric that accounts for the fact that ZAP-70 localises to the cytoplasm and becomes membrane-bound upon receptor triggering.

Specifically, we introduced the computation of *membrane-bound ZAP-70*, defined as the difference between the signal measured at membrane regions contacting the target cell and the signal at membrane regions not contacting the target. This subtraction was performed separately for protrusion membranes and for main body membranes, as these regions differ in apparent brightness and protrusions look considerably dimmer. Negative values resulting from background subtraction were set to zero.

To determine whether ZAP-70 recruitment was higher in protrusions than in the main cell body, an additional normalization step was required. Once membrane-bound, ZAP-70 will appear brighter in protrusions due to the higher membrane-to-pixel ratio. We therefore normalized membrane-bound ZAP-70 to the enrichment of the uniformly distributed membrane marker SNAP-CAAX obtaining the *relative membrane bound ZAP-70* parameter. *Relative membrane-bound ZAP-70* was consistently higher in protrusions than in the main body membrane (Two new panels (B,C) were added to Fig. 3).

A detailed explanation of this quantification is now included in the Methods section. Figure 3 has been changed to include this new quantification (in B,C). Because of the addition of two panels and due to editorial requirements on figure size, we have now split Figure 3 in two figures:

Figure 3: CAR activation is enhanced at protrusions

Figure 4: LAT clusters form at protrusion contacts

Both figures are still discussed in the same Results section “CAR clusters in protrusions are activation hotspots”, that has also been changed to include text changes about the new quantification parameters (lines 9-15, page 6). The graphs in the new Figure 8C-D (summary figure) have been changed to include the new ZAP-70 data.

- An additional and potentially informative metric for assessing signalling efficiency at microvilli would be the ratio of ZAP-70-HALO intensity to CAR-GFP intensity. If microvilli are indeed privileged sites for initiating signalling, one would expect this ratio to be higher at microvilli compared to the cell body.

We thank the reviewer for this great suggestion. We have computed the ratio of membrane-bound ZAP-70-to-HER2-CAR within CAR clusters (detected as size-constrained accumulations of CARs whose mean signal is at least twice higher from the mean CAR signal at the plasma membrane mask) located both at protrusions and at the main body membranes. We have performed this analysis on CAR clusters to focus on the analysis of CARs engaging their ligands. The result (shown in Fig. 3D) indicates that CAR clusters located at actin-rich protrusions recruit significantly more ZAP-70. Enhanced ZAP-70-to-CAR ratio at protrusion clusters suggests that protrusions do not only aid signalling through an enhanced accumulation of receptors, but through a mechanism that allows that the accumulated receptor is more efficiently triggered. We consider that this suggestion expanded our model and gave better mechanistic insights. We have now included the quantification in Fig. 3D and made changes to the text (lines 15-19, page 6). We have also expanded the discussion (lines 44-46, page 8) Additional information about CAR cluster segmentation is now included in the Methods section.

This suggestion prompted us to perform a similar analysis for the CD45 dataset. The analysis revealed that CAR clusters at protrusions contain a significantly lower amount of CD45 as

shown by a lower CD45-to-CAR ratio (new panel D in Figure 6). This correlates the increased CAR activation with an increased CD45 exclusion and allowed us to suggest that a more efficient kinetic segregation might be behind the privileged ability of actin-rich protrusions to not only accumulate, but also activate, CAR receptors. We have expanded the results (lines 21-22, page 7) and discussion (lines 8-12, page 9) sections accordingly.

- A potentially important caveat in the quantification of LAT clusters at microvilli versus the cell body (Fig. 3E) is the effect of optical sectioning. For microvilli, the imaging plane is likely to capture the entire membrane-associated signal, whereas a CAR cluster on the flatter cell body may be associated with a LAT cluster that lies outside the z-plane. This could contribute to the relatively modest, though statistically significant, difference reported.

We hope we understand the reviewer's concern. We wanted to clarify, that we are analysing LAT enrichment at contacts, not throughout the entire in focus protrusion (Figure 4B and C). Therefore, the quantification informs on the amount of LAT in the very proximity of CAR clusters formed at contacts. The quantification thus excludes signal derived from clusters segregating from the contact also for protrusive membranes.

Minor comment:

- I couldn't find the supplementary movies in the files for reviewers so I could not evaluate these.

We apologise and hope supplementary movie files will be available upon submission of a revised manuscript.

Non-essential suggestion:

- There are freely available machine-learning approaches to segmenting cells, such as Ilastik (<https://www.ilastik.org/>), that may allow more nuanced segmentation of different types of protrusions (eg microvilli, filopodia and pseudopodia). It might be informative to make a distinction between the distribution of CAR, Lck and CD45 in the three types.

We thank referee 2 for the recommendation and will try Ilastik out for future research projects.

Dear Francesca,

Thank you for submitting your revised manuscript to The EMBO Journal. It has now been seen by both original reviewers, and I have attached their comments below. As you can see, they are generally satisfied with the performed revisions, while reviewer #2 indicates that one of their initial concerns was not sufficiently addressed.

Therefore, I would like to invite you to address the remaining concern by reviewer #2, as well as the editorial and formatting points below before I can extend official acceptance of the manuscript:

1. As part of the EMBO Press transparent editorial process, The EMBO Journal will publish online a Peer Review File to accompany accepted manuscripts. This file will be published in conjunction with your paper and will include the anonymous referee reports, your point-by-point response and all pertinent correspondence relating to the manuscript, including decision letters. Please note that the Author Checklist will be published at the end of the Peer Review File. Please let us know if you want to remove or not any figures or data from the Peer Review File prior to publication. Please note that retaining unpublished data in the Peer Review File means that these count as published and that the Peer Review File would need to be referenced in future publications.
2. In the provided author checklist, please clarify the information in the lane 81: it is marked as "not applicable", while column C indicates where this information is available in the study.
3. Please check that the funding information is correct and identical both in the manuscript and our online system. Currently, Gabrielle's Angel Foundation Medical Research Award and the Pershing Square Sohn Prize for Young Investigators in Cancer research are not listed in our online system.
4. Please add the label "Abstract" to the corresponding section.
5. CRediT has replaced the traditional author contributions section because it offers a systematic, machine-readable author contributions format that allows for more effective research assessment. Please remove the Authors Contributions from the manuscript and use the free text boxes beneath each contributing author's name in our online submission system to add specific details on the author's contribution. More information is available in our guide to authors.
6. Please rename "Competing Interests Statement" into "Disclosure and competing interests statement".
7. Please update the references according to The EMBO Journal style - where there are more than 10 authors on a paper, the first 10 should be listed, followed by 'et al.'
8. In the legend for Figure 6, please check the labelling of the panels - the panels E, F seem to be mislabelled as F, G.
9. In the legend for figure EV2, the panels D, E, F, G appear mislabelled as E, F, E, F. Please check and correct.
10. Please upload Tables EV1-3 as single files and adjust the nomenclature within the files to Table EV1 etc.
11. Please zip each individual movie file together with their corresponding legend provided in a text file.
12. In the Appendix, please add a cover page containing a title stating "Appendix for 'MS title'" and Table of Contents with each item listed and the corresponding page number. Please update the nomenclature to Appendix Figure S1, etc. Please rotate the figure to fit a portrait format.
13. During our standard source data check, I noticed unexplained numerical repetitions in the source data for several figures. I have attached the corresponding files with the detected duplications labelled in colour. In the source data for figures 5B and 2C, please pay attention to the blocks of repetitive values in two of the columns. Please take a look and correct if needed. A brief explanation would be very helpful - I appreciate that these duplications can also occur due to specific measurement or calculation methods used.
14. In our standard image integrity check, we have noticed image reuse between figures 4A and 5A, the lower row, second panel. Please check and correct, especially since these images appear to represent different experimental conditions.
15. Our data editors have flagged the following issues in figure legends that need correcting:
 - Please provide information on the number and nature of replicates in the legends of figures 1C, 4C, 6C, D; 8B-D; EV2 A.
 - Please define the yellow arrow in the legend of figure 6E.
 - Please define the yellow and red arrows in the legend of figure 6E.
16. Papers published in The EMBO Journal are accompanied online by a 'Synopsis' to enhance discoverability of the manuscript. It consists of A) a short (1-2 sentences) summary of the findings and their significance, B) 3-4 bullet points highlighting key results and C) a synopsis image that is 550x300-600 pixels large (width x height, jpeg or png format). You can either show a model or key data in the synopsis image. Please note that the image size is rather small and that text needs to be readable at the final size. Please send us this information together with the revised manuscript.

Thank you again for giving us the chance to consider your manuscript for The EMBO Journal. I look forward to receiving the final version!

With kind regards,

leva

We realize that it is difficult to revise to a specific deadline. In the interest of protecting the conceptual advance provided by the work, we recommend a revision within 3 months (28th May 2026). Please discuss the revision progress ahead of this time with the editor if you require more time to complete the revisions.

Referee #1:

The authors have fully addressed my concerns regarding the normal functionality of the TCR signal transduction pathway in the presence of the labeled proteins discussed. The controls presented adequately demonstrate that the fundamental elements of the TCR signal transduction pathway are not affected by the inclusion of Halo-tagged LAT, Zap-70, Lck, or CD45.

I appreciate the authors' efforts to clarify the details of their imaging techniques and analysis. While the authors acknowledge the limitations of single-plane imaging, I feel that the details the authors have now provided are sufficient to allow reasonable interpretation of their results. I also feel that the authors' determination of pre-contact and post-contact sites is now clearer and the terminology is more accurate. Lastly, I also believe the authors' specifying that imaging began prior to contact is particularly helpful to the interpretation of their observations.

The authors' additional extended data and additional citations now listed fully address my concerns regarding the functionality of the tagged constructs.

I appreciate efforts the authors have made to expand on previous studies using CAR T cells. I believe that the added discussion helps put the authors' current work in the context of the broader field of TCR complex organization and signaling dynamics. I also feel that connecting their observations to the rapidly evolving field of CAR signaling dynamics adds a significant therapeutic relevance that enhances the impact and broadens the interest of their manuscript.

The authors' additional discussion addresses my concerns regarding any possible discrepancy between their observations and known TCR signaling component dynamics. This discussion also enhances the impact of the manuscript by more fully integrating the above observations into the broader field of CAR signaling dynamics.

In terms of the Discussion, I feel that the authors now more fully integrate their observations with the broader field of CAR signaling dynamics.

I acknowledge the authors' justification for their use of primary CD4+ T cells. I also agree that their choice is relevant given their model system and that their results clearly and adequately demonstrate that the processes observed are present in primary cells and are not an artifact of the cell line.

I appreciate the authors' efforts to enhance the clarity of the statement in paragraph 4 of the Discussion, where they state that the "primary mechanism by which actin-rich protrusion (sic) promote increased CAR accumulation..." is through the exclusion of CD45, and believe the text now more accurately reflects their observations as well as published literature.

In summary, I think the paper is ready to publish.

Referee #2:

I thank the authors for taking on board my comments, and am satisfied that the changes they made have address all concerns except for one where there may have been some confusion about and which I have clarified below:

To clarify my comment about LAT in out of focus areas of the cell body (Fig 4):

LAT condensates associated with CAR clusters are often slightly spatially offset (as is evident in Fig 4A). For microvilli there is unlikely much signal from outside the optical section and thus all the LAT recruited would be visible, while in contacts involving the cell body (in a synapse) the membrane around the CAR clusters extends axially above and below the optical section, and thus some of the LAT condensate associated with the activated CAR might be outside the focal plane.

If not already done, volumetric (z-integrated) quantification or orthogonal views of representative contacts would help confirm that the reported differences are not influenced by differences in axial geometry.

Answers to Referee 2:

I thank the authors for taking on board my comments, and am satisfied that the changes they made have address all concerns except for one where there may have been some confusion about and which I have clarified below:

To clarify my comment about LAT in out of focus areas of the cell body (Fig 4): LAT condensates associated with CAR clusters are often slightly spatially offset (as is evident in Fig 4A). For microvilli there is unlikely much signal from outside the optical section and thus all the LAT recruited would be visible, while in contacts involving the cell body (in a synapse) the membrane around the CAR clusters extends axially above and below the optical section, and thus some of the LAT condensate associated with the activated CAR might be outside the focal plane.

If not already done, volumetric (z-integrated) quantification or orthogonal views of representative contacts would help confirm that the reported differences are not influenced by differences in axial geometry.

We thank the Referee for further clarifying their concerns.

We agree that LAT clustered in the main body membrane may be able to diffuse in and out of the focal plane more easily than in protrusions.

We have changed the text to reflect this limitation in the Results and Discussion sections (highlighted in yellow).

Lines 31-35, Page 6 *"It is important to note that segregation of LAT clusters from contacts could affect the quantification of LAT relative enrichment at protrusions and main body membrane differently, as clusters on the cell body may be able to more easily diffuse away from the imaged plane."*

Lines 4-7, Page 9 *"Additionally, we could observe LAT clusters segregation from protrusion contacts. It is possible that such events could occur at clusters on the main body membrane and may have been missed because of our 2D imaging modality."*

We would like to highlight that the quantification presented in Figure 4B-C refers to LAT relative enrichment at contacts which only include the small area of the protrusions making contact and also does not account for the segregating LAT clusters.

We nevertheless chose to include the data on segregating LAT clusters in the manuscript, as it represents an intriguing observation that we hope will be explored further. But as the referee suggested, no conclusions can be made on the segregating clusters from our 2D imaging approach.

Dear Francesca,

Thank you for incorporating the final editorial and formatting requests in the manuscript. I apologise for the slow process from our side due to the large number of revised submissions that we currently receive. I am now pleased to inform you that your manuscript has been accepted for publication. Congratulations with a nice study!

Before we forward your manuscript to our publishers, we would like to propose some edits in the manuscript title, abstract and synopsis. I have also written a short blurb that will accompany the title of your manuscript in our online table of contents. Please take a look at the proposed text changes in the attached file and let me know if any corrections are needed.

You may qualify for financial assistance for your publication charges - either via a Springer Nature fully open access agreement or an EMBO initiative. Check your eligibility: <https://link.springer.com/journal/44318/how-to-publish-with-us>

If you have any questions, please do not hesitate to contact the Editorial Office or me directly. Thank you for this interesting contribution to The EMBO Journal!

Best wishes,

leva

leva Gailite, PhD
Senior Scientific Editor
The EMBO Journal
Meyerhofstrasse 1
D-69117 Heidelberg
Tel: +4962218891309
i.gailite@embojournal.org